

# Integration of geological uncertainty into geophysical inversion by means of local gradient regularization

Jeremie Giraud[1], Mark Lindsay[1], Vitaliy Ogarko[2], Mark Jessell[1], Roland Martin[3], and Evren Pakyuz-Charrier[1]

[1] Centre for Exploration Targeting, School of Earth Sciences (University of Western Australia), 35 Stirling Highway WA Crawley 6009, Australia.

[2] The International Centre for Radio Astronomy Research, 7 Fairway, Ken and Julie Michael Building, The University of Western, Australia, WA Crawley 6009, Australia.

[3] Géoscience Environnement Toulouse, Observatoire Midi-Pyrénées, 14 Avenue Edouard Belin, 31400 Toulouse, France.

*Correspondence to*: Jérémie Giraud (jeremie.giraud@research.uwa.edu.au)

**Abstract.** We introduce a workflow integrating geological uncertainty information in order to constrain gravity inversions. We test and apply this approach to data from the Yerrida Basin (Western Australia), where we focus on prospective greenstone belts beneath sedimentary cover. Geological uncertainty information is extracted from the results of a probabilistic geological modelling process using geological field data and their uncertainty as input. It is utilized to locally adjust the weights of a minimum-structure gradient-based regularization function constraining geophysical inversion. Our results demonstrate that this technique allows geophysical inversion to update the model preferentially in geologically less certain areas. It also indicates that inverted models are consistent with both the probabilistic geological model and geophysical data of the area, reducing interpretation uncertainty. The interpretation of inverted models finally reveals that the recovered greenstone belts may be shallower and thinner than previously thought.

## 1 Introduction

The integrated interpretation of multiple data types and disciplines in geophysical exploration is a powerful approach to mitigating the limitations inherent to each of the datasets. For instance, gravity data, which has poor horizontal resolution, can be integrated with seismic inversion to mitigate the poor lateral resolution of seismic inversion (Lelièvre et al., 2012). Likewise, geological modelling and geophysical inversions are routinely performed in the same area to obtain an subsurface model consistent with geological and geophysical measurements (Guillen et al., 2008; Lelièvre and Farquharson, 2016; Pears et al., 2017; Williams, 2008). When sufficient prior information is available, petrophysical constraints can be derived for inversion (Lelièvre et al., 2012; Paasche and Tronicke, 2007), and integrated with geological modelling to derive local constraints (Giraud et al., 2017). However, in exploration scenarios, this might be impractical as the available petrophysical information may be insufficient to allow us to derive such constraints (Dentith and Mudge, 2014). In such cases, when more than one





geophysical dataset is available, practitioners may rely on joint inversion using structural constraints (e.g., Gallardo and Meju, 2003; Haber and Oldenburg, 1997; Zhdanov et al., 2012).

Alternatively, when one of the datasets has a spatial resolution that is superior to the others, structural information can be transferred into the gradient regularization constraint for the inversion of the lesser resolving method(s), thus mitigating some

of the challenges faced by joint inversion in such cases into what (Brown et al., 2012) termed guided inversion. This strategy has been applied in recent years using the interpretation of predominantly propagative data (e.g., seismics, ground-penetrating radar) to constrain the inversion of diffusive data (e.g., diffusive electromagnetic methods), see (Yan et al., 2017) and references reported therein. However, this avenue remains relatively unexplored to date.

In this article, we broaden the applications of guided inversion and explore the integration of non-geophysical information in

inversion, such as geological uncertainty, into what we call uncertainty-guided inversion where we focus on the complementarity of information content between the datasets. We introduce a new technique that integrates local uncertainty information derived from probabilistic geological modelling in the inversion of potential field data, following recommendations of (Jessell et al., 2014, 2010; Lindsay et al., 2013; Lindsay et al., 2014; Wellmann et al., 2014, 2017). In contrast to (Giraud et al., 2016, 2017) who derives local petrophysical constraints from petrophysical measurements and

geological modelling results, constraints used in uncertainty-guided inversion are based solely on the local conditioning of a gradient regularization function, thereby offering the possibility to integrate probabilistic geological modelling into geophysical inversion in the absence of sufficient petrophysical information. This conditioning relies on the calculation of local weights derived from prior geological information. In this study, we utilize a probabilistic geological model (PGM) (Pakyuz-Charrier et al., 2018)consisting of the observation probability of the different lithologies of the area in every model

cell. More specifically, we utilize the information entropy (Shannon, 1948; Wellmann and Regenauer-Lieb, 2012), which measures geological uncertainty in probabilistic models. We calculate it in each model cell of the PGM to derive spatially varying weights applied to the gradient regularization function used during inversion.

The integration methodology we develop is similar in philosophy to (Brown et al., 2012; Guo et al., 2017; Wiik et al., 2015), who extract continuous structural information from seismic data to adjust the strength of the regularization term locally in

order to promote specific structural features during electromagnetic inversion. However, our work differs from these authors in four main respects. Firstly, the geophysical problem we tackle is different in nature as we constrain potential field data in hard rock scenario instead of electromagnetic data in soft rock studies. Secondly, we use a metric encapsulating geological uncertainty derived from geological measurements, whereas, in contrast, previous studies use other geophysical attributes. Thirdly, we allow inversion to update the model preferably in the most uncertain parts of the geological model, instead of

encouraging a certain degree of structural similarity between two geophysical inverse models. Finally, while previous work involve mostly 2D models, every step of our modelling is performed purely in 3D.



In this paper, we introduce the methodology and field application as follows. In the methodology Section, we first introduce the inversion and integration scheme and algorithm, and provide essential background information about probabilistic geological modelling. We then provide the essential background about information entropy before detailing its usage in inversion. In the ensuing section, we present a field application case focused on the Yerrida Basin (Western Australia), starting

with the introduction of the geological context and modelling procedure. We then analyse the influence of local regularization conditioning on inverted models and demonstrate how it allows clearer and more reliable interpretation of the buried greenstone belts than when it is not utilized.

## 2    Modelling procedure

### 2.1    Inversion methodology

The inversion procedure we propose integrates spatially varying prior information to weight the regularization function locally (e.g., in each cell). It is implemented in an expanded version of the least-square inversion platform Tomofast-x (Martin et al., 2013, 2018), which offers the possibility to condition the regularization function (Tikhonov and Arsenin, 1977) of (Li and Oldenburg, 1996) locally using geological uncertainty. This is achieved by incorporating prior information into a structure-based regularization function in a fashion similar to (Brown et al., 2012; Wiik et al., 2015; Yan et al., 2017) by locally adjusting

the related weight.

Solving the inversion problem regularized in this fashion consists of finding a model $\boldsymbol{m}$ that minimizes the objective function $\theta$ given below:

$$\theta(\boldsymbol{d},\boldsymbol{m}) = \underbrace{\left\|\boldsymbol{W_d}\big(\boldsymbol{d}-\boldsymbol{g}(\boldsymbol{m})\big)\right\|_2^2}_{\text{Data term}} + \underbrace{\left\|\boldsymbol{W_m}\big(\boldsymbol{m}-\boldsymbol{m_p}\big)\right\|_2^2}_{\text{Model term}} + \underbrace{\alpha\|\boldsymbol{W_H}\nabla\boldsymbol{m}\|_2^2}_{\text{Structural regularization term}}, \tag{1}$$

where $\boldsymbol{d}$ relates to the geophysical measurements to be inverted, $\boldsymbol{g}$ is the forward modelling operator; $\boldsymbol{m}$ relates to the model

being searched for, and $\boldsymbol{m_p}$ is the prior model; $\boldsymbol{W_d}$, $\boldsymbol{W_m}$ and $\boldsymbol{W_H}$ are diagonal weighting matrices corresponding to data noise, model weighting and gradient regularization, respectively. The model term is a ridge regression constraint term (Hoerl and Kennard, 1970).

The structural regularization term in Eq. (1) enforces structural constraints during inversion. It is weighted locally by matrix $\boldsymbol{W_H}$, which can be derived from prior information (see Subsect. 2.3 for details). The positive free parameter $\alpha$ controls the

overall weight of the regularization term; $\nabla$ is the spatial gradient operator. Note that $\|\nabla\boldsymbol{m}\|_2$, estimates the amount of structure in inverted physical property model $\boldsymbol{m}$. Note that parts of the model where $\boldsymbol{W_H}=0$ are excluded from the calculation of the structural regularization and can change freely to accommodate geophysical data.



## 2.2 Probabilistic geological modelling

Probabilistic geological modelling is performed using the Monte-Carlo Uncertainty Estimator (MCUE) method of (Pakyuz-Charrier et al., 2018), which is an uncertainty propagation method for 3D implicit geological modelling using geological rules and geological orientation measurements (foliation and interface) as inputs. The sampling algorithm perturbs orientation data by sampling probability distributions describing the uncertainty of orientation data to produce a series of unique altered models. Realizations that do not respect a series of geological rules are considered implausible and are rejected. Coupled to the 3D geological modelling engine of Geomodeller© (Calcagno et al., 2008), it produces a set of plausible geological models representing the geological model space (Lindsay et al., 2013b). The observation probabilities constituting the probabilistic geological model (PGM) are obtained, in each model cell, by calculating the relative observation frequencies of the different lithologies from the set of geological models. For the $i^{\text{th}}$ model cell of a PGM containing $L$ lithologies, vector $\boldsymbol{\psi^i} = \left[\psi_{k=1}^i, \dots, \psi_{k=L}^i\right]$ contains the observation probabilities of each lithology. As we show in the next subsection, the resulting PGM can be used to estimate uncertainty levels and as a source of prior information.

## 2.3 Utilisation of information entropy for local conditioning

Information entropy has recently been introduced in geological modelling by (Wellmann and Regenauer-Lieb, 2012) and is gaining popularity as a measure of uncertainty in probabilistic geological modelling (de la Varga et al., 2018; de la Varga and Wellmann, 2016; Lindsay et al., 2014; Lindsay et al., 2013; Pakyuz-Charrier et al., 2018; Schweizer et al., 2017; Thiele et al., 2016; Wellmann et al., 2017; Yamamoto et al., 2014). Quoting (Schweizer et al., 2017), information entropy "quantifies the amount of missing information and hence, the uncertainty at a discrete location". For the $i^{\text{th}}$ model-cell, it is given as (Shannon, 1948):

$$H^i = H(\boldsymbol{\psi^i}) = -\sum_{k=1}^{L} \psi_k^i \log\left(\psi_k^i\right). \tag{2}$$

Instead of using $\boldsymbol{H}$ directly, we calculate $\boldsymbol{W_H}$ utilising its normalized complementary, which reflects the degree of certainty across the model. Let us express $\boldsymbol{W_H}$ as follows, for the $i^{\text{th}}$ model cell:

$$W_H^i = \frac{\max \boldsymbol{H} - H^i}{\max \boldsymbol{H} - \max \boldsymbol{H}} \tag{3}$$

The consequence of Eq. (2) and 3 is that $\boldsymbol{W_H}$ is minimum at interfaces and in areas poorly constrained by geological information, and equal to unity in areas where the geology is well resolved. Consequently, the conditioning process attaches small weights to the structural term of Eq. (1) in uncertain cells, while consistently high values will be applied to the most geologically certain cells. As a result, it enables the inversion algorithm to favour nearly constant changes in the inverted model in contiguous certain groups of cells (e.g., where $\boldsymbol{W_H} \rightarrow 1$) while relaxing the regularisation constraints in the most uncertain parts (e.g., where $\boldsymbol{W_H} \rightarrow 0$).

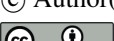



For proof-of-concept validation, we simulated an idealized case study to assess the capability of inversion using $W_H$ as per Eq. (3) to improve inversion results compared to the non-conditioned case (e.g., with $W_H$ equal to the identity matrix). We tested the proposed methodology using synthetic geophysical data calculated from the structural geological model of the

Mansfield area of (Pakyuz-Charrier et al., 2018), which we populated in the same fashion as (Giraud et al., 2017). The analysis of inverted models demonstrates the potential of the proposed inverse modelling scheme to ameliorate inversion results and to reduce interpretation uncertainty (see details in Appendix A). Importantly, in this synthetic case, local conditioning allows geophysical inversion to significantly improve the imaging of geologically uncertain areas. From the success of that theoretical proof-of-concept study, we are confident that our integration method can be tested here using real world data for field

validation.

## 3 Field validation: Yerrida Basin case study

### 3.1 Geological context and geophysical survey setup

The Yerrida Basin is located in the southern part of the Capricorn Orogen, at the northern margin of the Yilgarn Craton in Western Australia (Fig. 1a), and extends approximately 150km N-S and 180 km E-W (Fig. 1b). The studied area is delimited

in the northwest by the Goodin Fault, which represents a faulted contact between the Yerrida Basin and the Bryah-Padbury Basin. The structures of interest in this work are: Archean greenstone belts extending north-northwest that are unconformably overlain by Paleoproterozoic sedimentary rocks the form the Yerrida Basin. Features A and B (Fig. 1a and Fig. 1b) indicate the interpreted position of buried Wiluna Greenstone Belt. Where the Wiluna Greenstone Belt is exposed, it is host to base and precious metal mineralisation (Williams, 2009). With a relatively high positive density contrast (expected to be between 190

and 270 kg.m$^{-3}$) to the Yilgarn Craton granite-gneiss host, mafic greenstone belts A, B, and C are suitable targets for gravity inversion. Interpretations from multiple studies in the region, e.g, (Johnson et al., 2013; Pirajno et al., 1998; Pirajno and Adamides, 2000; Pirajno and Occhipinti, 2000) were compiled while additional geological measurements acquired in 2015, 2016 and 2017 complemented legacy data (Occhipinti et al., 2017; Olierook et al., 2018). This allowed the revision of existing models and improved our understanding of the area. This, in turn, also highlights the challenges presented by the

characterization of greenstone belts A, B and C, and that further geophysical analysis through constrained inversion is a useful pathway for reducing exploration risk.

Inverted geophysical data consists of ground surveys obtained from Geoscience Australia (http://www.ga.gov.au/data-pubs). Post-processing includes spherical-cap and terrain corrections and the removal of the regional trend to obtain the complete Bouguer anomaly, which we forward model following (Boulanger and Chouteau, 2001). As most data points were acquired 1

to 4 km apart, the dataset was resampled to match the geological model discretization, making up a total of 4882 measurement points. The model is discretized into $100 \times 100 \times 42$ cells of dimensions 2.335 km $\times$ 1.875 km $\times$ 1.0475 km, down to a depth



of 44 km, making up a total of 420000 cells. We utilize the integrated sensitivities technique of (Li and Oldenburg, 2000; Portniaguine and Zhdanov, 2002) to precondition the data term in Eq. (1) in order to balance the decreasing sensitivity of gravity field data with depth.

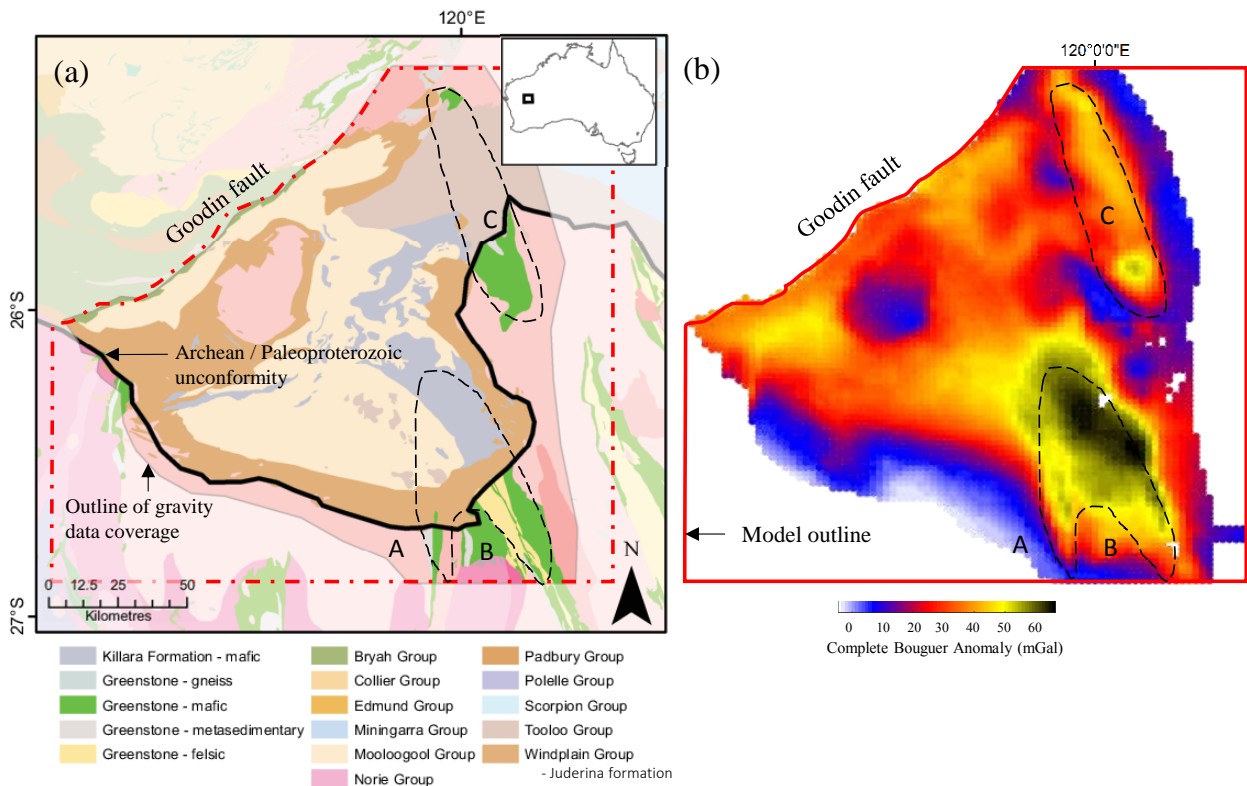

**Figure 1.** Geological context and geophysical data. (a) Geological map of the area and (b) complete Bouguer anomaly. The dashed lines delineate the possible sub-basin extent of the mafic greenstone belts A, B and C.

### 3.2 Geological modelling

Geological data consists of in-situ structural measurements (interfaces and foliations) and interpretation of aeromagnetic, airborne electromagnetic, Landsat 8 and ASTER data. Legacy data from the Geological Survey of Western Australia (Pirajno and Adamides, 2000) and CSIRO (Ley-Cooper et al., 2017) were used, to which about 600 data points and petrophysical measurements from recent geological field campaigns were added. Although the available petrophysical measurements are not used to derive petrophysical constraints because of the insufficient sampling of several of the modelled lithologies, they are a useful source of information to populate geological models and for interpretation.

These datasets were used jointly to build a reference geological model for MCUE simulations, after which lithologies with similar density contrasts were merged and subsequently treated as a single rock type. Uncertainty related to structural





measurements was subsequently used as inputs to the MCUE perturbations (Pakyuz-Charrier et al., 2018) of the reference model to generate a collection of 500 accepted models. Information extracted from the PGM displayed in Fig. 2. Figure 2a shows the lithologies with the highest probability for each cell of the PGM. The associated $W_H$ values used in inversion are shown in Fig. 2b. The starting model for inversion, which we also use as prior model $m_p$, is equal to the mean model of the 500 plausible models generated by MCUE, is shown in Fig. 2c.



**Figure 2.** Geological modelling results. (a) Most probable lithology in each model cell (same colour code as in Fig. 1) (b) values used for local regularization conditioning, (c) and starting model derived from PGM and prior petrophysical information). In (a), "background" refers to all the lithologies that have a density contrast equal to 0 kg.m⁻³.



## 3.3 Inversion results and analysis

Our analysis aims at determining the influence of the local conditioning of structural constraints on inversion through comparison with the non-conditioned case, all other things remaining constant.

### 3.3.1 Comparative analysis strategy

Prior to examination of the inverted models, we analyse geophysical data misfit after inversion for a fixed number of major iterations (100) of the least-square geophysical inverse solver superior to that needed for convergence of the inversion algorithm (~10 in this case). This enables us to ensure that the inversion results we compare produce, in our case, similar gravity anomalies. Our study of inverted models focuses on results obtained through usage of non-conditioned (Fig. 3a) and conditioned regularization function (Fig. 3b) using $W_H$ (Fig. 2b). In addition to departures from the starting model, variations

between the two cases are studied by visual comparison of Fig. 3a and Fig. 3b, through qualitative (Fig. 3c) and quantitative comparative analysis (Fig. 3d-e). Our interpretation of inversion results is complemented by metrics quantifying the differences between models. We give particular attention to model cells where the probability of mafic greenstone is superior to zero. For these cells, we classify lithologies by identifying cells with a density contrast corresponding to mafic greenstone.

### 3.3.2 Results

Data root-mean-square (RMS) error decreases during inversion from 12.46 mGal to reach 1.59 mGal and 1.53 mGal for the non-conditioned and conditioned cases, respectively. The corresponding data misfit maps show a linear correlation coefficient of 0.999 (see details in Appendix A3 and Fig. A3). This similarity illustrates that, as in many other studies, most changes related to holistic data integration in geophysical inversion occur primarily in model space, hence reducing the effect of non-uniqueness (Abubakar et al., 2012; Brown et al., 2012; Demirel and Candansayar, 2017; Gallardo et al., 2012; Gallardo and

Meju, 2004, 2007, 2011; Gao et al., 2012; Giraud et al., 2017; Guo et al., 2017; Heincke et al., 2017; Jardani et al., 2013; Juhojuntti and Kamm, 2015; Molodtsov et al., 2013; Moorkamp et al., 2013; Rittgers et al., 2016; Sun and Li, 2016, 2017).

Qualitatively, comparison of Fig. 3a and Fig. 3b reveals that departures from the starting model (Fig. 2c) are more significant in the most geologically uncertain areas. Quantitatively, the RMS model update for cells characterized by $0 \leq W_H < 0.05$ (most uncertain group) is equal to 40.1 $kg/m^3$ and 51.5 $kg/m^3$, for the non-conditioned and conditioned cases, respectively,

whereas the same quantities are equal to 17.7 $kg/m^3$ and 16.9 $kg/m^3$ for the cells identified by $0.95 < W_H \leq 1$ (most certain group). This suggests that local regularization conditioning allows inversion to update the model preferentially in geologically uncertain areas. In turn, differences with the starting model in more geologically certain areas are reduced compared to the non-conditioned case. This effect of conditioning is corroborated by Fig. 3c where the longest distances to the dashed line, which represents equal model update for the two studied cases, occur in geologically uncertain areas. This also

translates in higher difference between model updates of the two compare cases in Fig. 3d for lower values of $W_H$. In addition,





we observe that local conditioning produces stronger density contrasts in Fig. 3b in some of the areas where the conditioning values are higher in Fig. 2b. Furthermore, structures in the inverted model are easier to identify when local conditioning is used. It is confirmed by global roughness measures $\|\nabla m\|_2$ equal to 3.4 $(kg/m^3)/m$ and 4 $(kg/m^3)/m$ for the non-conditioned and conditioned cases, respectively. More specifically, as shown by Fig. 3e, this difference arise in parts of the
5   model associated with lower $W_H$, which characterize uncertain areas, including interfaces between lithologies.

The recovered greenstone belts are shown in Fig. 3a and Fig. 3b. In Fig. 3b, the extension of recovered mafic greenstone belts is significantly different than when geological uncertainty is not accounted for (Fig. 3a). In particular, belt A is significantly larger in Fig. 3b than in Fig. 3a ($2.4\times10^2$ km$^3$ vs $4.6\times10^2$ km$^3$). Similarly, the extent of belt C is increased overall (volume of $5.3*10^2$km³ vs $14\times10^2$km$^3$), while its different portions reconnect; the northern half is also significantly shallower and broader than in Fig. 2a and Fig. 3a. It appears that belt A remains thinner and shallower (Fig. 3b) than suggested by the preferred lithology volume (Fig. 2a). While similar geometries for belt B are recovered in Fig. 3a and Fig. 3b, they both differ from Fig. 2a as only the eastern part is preserved. Note that it is larger in Fig. 3b, with a volume 40% higher than in Fig. 3a. As discussed in the next subsection, these differences have a signification impact on the interpretation of inversions results and are important to understand the influence of local conditioning on inversion.





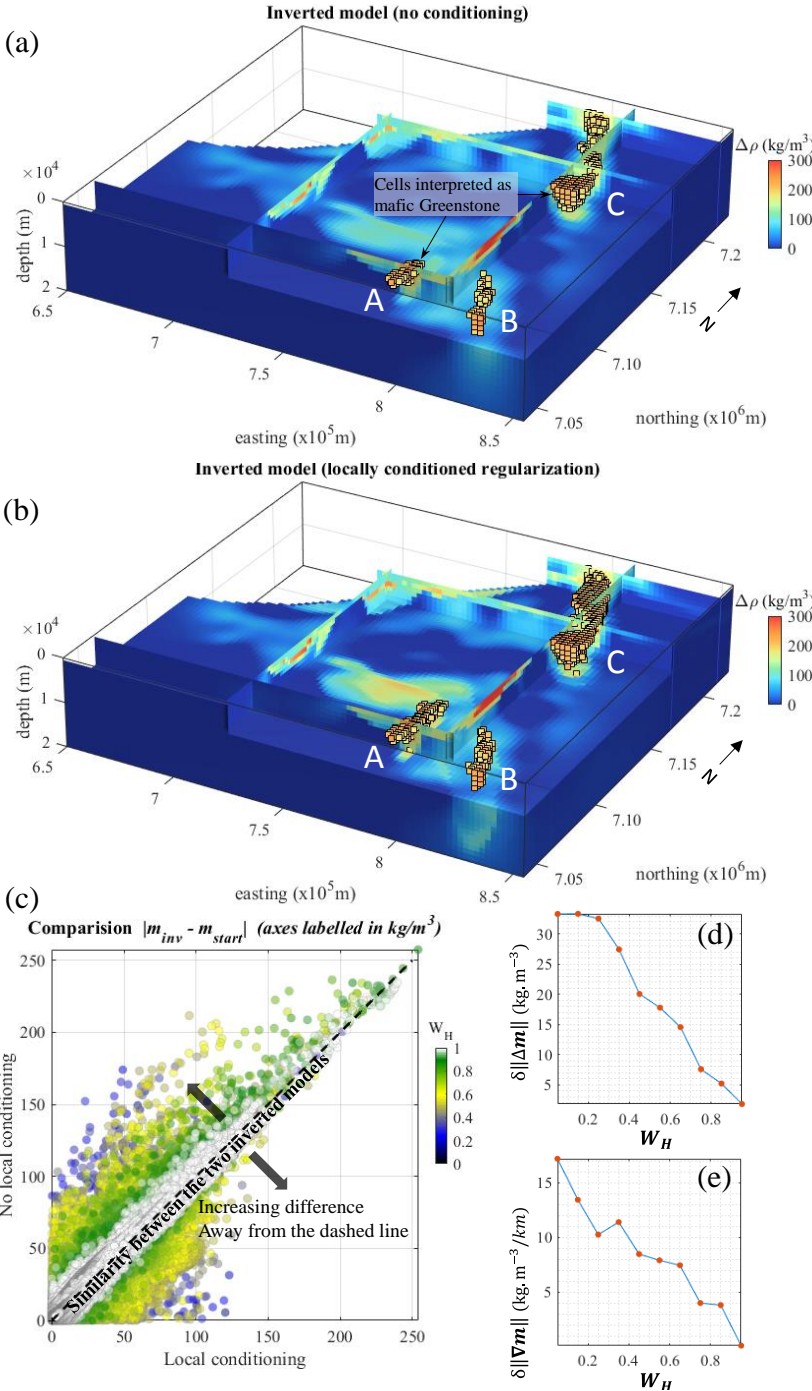

**Figure 3.** Comparison of inversion results. (a) inverted models with non-conditioned regularization weights, and (b) using local conditioning, (c) cross-plot between the corresponding absolute value of the update of the starting model, (d) difference in model updates $\delta\|\Delta m\| = \|m_{cond} - m_{nocond}\|_2$ as a function of values of $W_H$ and (e) difference in model roughness $\delta\|\nabla m\| = \|\nabla m_{cond}\|_2 - \|\nabla m_{nocond}\|_2$ as a function of values of $W_H$. The model cells labelled A, B and C are interpreted as mafic greenstone belts. All voxels are coloured as a function of density contrast.



### 3.4    Interpretation

Note that, in contrast to the differences between inversion results highlighted above for belts A and C, differences between the inverted models in the north-eastern part of the model and the different interpretations of belt B (Fig. 3a and Fig. 3b) are small. This shows that locally conditioned regularization does not enforce changes in the inverted model everywhere geological

uncertainty is high as uncertainty is only a reflection of potential errors. Rather, this indicates that in such cases, the guiding effect of such regularization will be exerted provided that it does not prevent the data term in $\theta(\boldsymbol{d}, \boldsymbol{m})$ as per Eq. (1) from decreasing. This also confirms that geophysical data is the main driver of the model updates in geologically uncertain areas. Instead of smooth departures from the starting model to match geophysical data regardless of geological considerations, local regularization constraints allow inversion to account for the probabilistic geological modelling of the area and for geological

uncertainty. It can therefore provide results that conform better to known geology.

In consequence, by confronting a probabilistic geological model encapsulating all MCUE realizations with geophysical measurements in an inversion scheme favouring model updates in the most geologically uncertain areas, inversion complements probabilistic geological modelling in that it guides and refines the interpretation of the geoscientific data in the area.

Geophysical inversion using geological uncertainty information (Fig. 2b) confirms the presence of high density anomalies that we interpret to be the mafic components of the greenstone as suggested by MCUE in several portions of the model. It also adjusts the outline and geometry of belts A, B and C to obtain a model honouring geological uncertainty information. In particular, mafic greenstone A and B may be smaller than the extent suggested by the PGM, and mafic greenstone C shallower than anticipated. Inversion results interpretation also reveal that greenstone B might be extended further to the east than

indicated by the preferred lithology volume (Fig. 2a) and that greenstone C may be thinner in its central part.

### 4    Conclusions

We have introduced a new integration scheme for the inversion of gravity data that utilizes a measure of geological uncertainty to calculate locally-conditioned gradient regularization constraints. Contrarily to previous work, this approach enables the integration of probabilistic geological modeling in geophysical inversion in the absence of petrophysical information sufficient

to the calculation of petrophysical constraints. It uses geophysical measurements to optimize the inverse problem by updating the physical property model preferably in geologically uncertain parts of the studied area during what we called *uncertainty-guided inversion*. This therefore partly mitigates inversion's non-uniqueness through the addition of constraints encouraging inversion to produce models that account for geological uncertainty across the entire inverted volume. We have demonstrated that it can be used collaboratively with geological modelling efficiently through field application in the Yerrida Basin.

Inversion results show that our integration methodology has the capability to refine the recovered physical property model and interpretations in portions of the model where geological uncertainty is high. Another advantage of the proposed technique





pertains to its time and cost-effectiveness as our workflow utilizes the PGM resulting from geological modelling and requires the same parameterization as non-conditioned inversion.

In the Yerrida Basin study area, application of the proposed methodology provided the effective delineation of the greenstone belts by quantitatively integrating geological measurements and geophysical data. Our findings suggest that some of the
greenstone belts covered by the basin might be shallower than previously anticipated and occupy smaller volumes. This is particularly pronounced in the North-East (belt C) where the resulting model is in agreement with the shallowest cases allowed by the PGM. Likewise, in the South (belt A), only the shallowest part of the mafic greenstone body can be resolved, while the south-eastern (belt B) greenstone belt appears to be limited in extension to the eastern part of the volume where it is the preferred lithology in the PGM. In such cases, this can also indicate that these greenstone bodies might be too thin to be imaged
by gravity data. These results have implications for the geological knowledge of the southern Capricorn Orogen as they indicate reduced (compared to the preferred lithology volume) mafic greenstone volumes under the Yerrida Basin on one hand, and decreased cover thickness on the other hand, thereby opening the door to updates in the geological interpretation of the history of the Yerrida Basin and potential new undercover exploration prospects.

Future research include the utilization of local petrophysical constraints of (Giraud et al., 2017) in the uncertainty-guided
inversion scheme we presented, as well as its application to weight the cross-gradient term of (Gallardo and Meju, 2003) in joint inversion schemes. With this last respect, uncertainty-guided inversion can be assisted in the most uncertain parts of the model by guided inversion (in the sense of Brown et al., 2012) or through cross-gradient joint inversion.

*Code and data availability.* Reference property models, inversion results and recovered models relating to the Yerrida Basin
shown in this article are made available online: Jeremie Giraud, Mark Lindsay, and Vitaliy Ogarko, 2018, Yerrida Basin Geophysical Modeling - Input data and inverted models. (Version version 1.0) [Data set]. Zenodo. http://doi.org/10.5281/zenodo.1238216. Reference property models, inversion results and recovered models relating to the synthetic case from the Mansfield area shown in this article are made available online: Jeremie Giraud, Vitaliy Ogarko, and Evren Pakyuz-Charrier, 2018, Synthetic dataset for the testing of local conditioning of regularization function using geological
uncertainty. (Version version 1.0) [Data set]. Zenodo. http://doi.org/10.5281/zenodo.1238529

## Appendix A: proof-of-concept using synthetic case study

This Appendix introduces the proof-of-concept of the proposed method through an idealized case study illustrating the potential of the proposed inverse modelling scheme to ameliorate inversion results and to reduce interpretation uncertainty. We use the same 3D density contrast model as (Giraud et al., 2017), which is obtained using the structural framework and
PGM of (Pakyuz-Charrier et al., 2018). The short presentation of the model below and the analysis of results provides essential information and support about the proof-of-concept of the methodology used in the paper.





## A1 Synthetic survey setup

The 3D unperturbed geological model was generated from contact (interface points) and surface orientation (foliations) field measurements collected in the Mansfield area (Victoria, Australia). It presents a Carboniferous mudstone-sandstone basin oriented N170, abutting a faulted contact with a folded ultramafic basement to the South-West. Model complexity was

5 artificially increased through the addition of a North-South fault and of a mafic intrusion.

The reference density contrast model (Fig. A4a) was obtained assigning density contrasts consistently with the structural setting of the unperturbed model, assuming a flat topography. Density contrasts of 0 and 100 kg.m⁻³ were assigned to the upper and lower basin lithotypes, respectively. Mafic rocks were assigned a density contrast of 200 kg.m⁻³ while the density contrast of the ultramafic basement was set to 300 kg.m⁻³.

10 MCUE perturbations of the reference model were performed using standard measurement uncertainty values recommended by metrological studies as reported by (Allmendinger et al., 2017; Novakova and Pavlis, 2017) generating a series of 300 models subsequently combined into a PGM. The resulting volume representing the $W_H$ values calculated from this PGM in each cell of the model as per Eq. (3) is show in Fig. A4b.

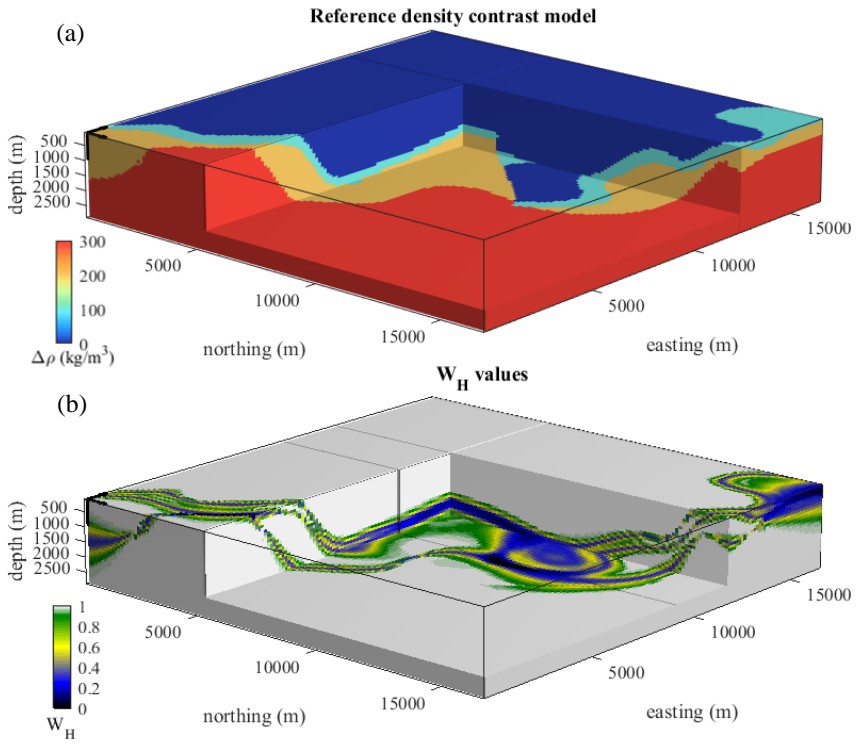

15 **Figure A4.** Reference model and $W_H$ values used for local regularization conditioning. (a) Unperturbed reference model with density contrast value, (b) uncertainty values used for local regularization conditioning.



## A2 Comparison of inversion results

To assess the impact of local conditioning of the regularization function, we compare inversions using non-conditioned (Fig. A5a) and locally conditioned (Fig. A5b) regularization function, respectively. Please note that, simulating the absence of prior petrophysical information, a homogenous starting model set to 0 kg.m⁻³ was used in both cases.

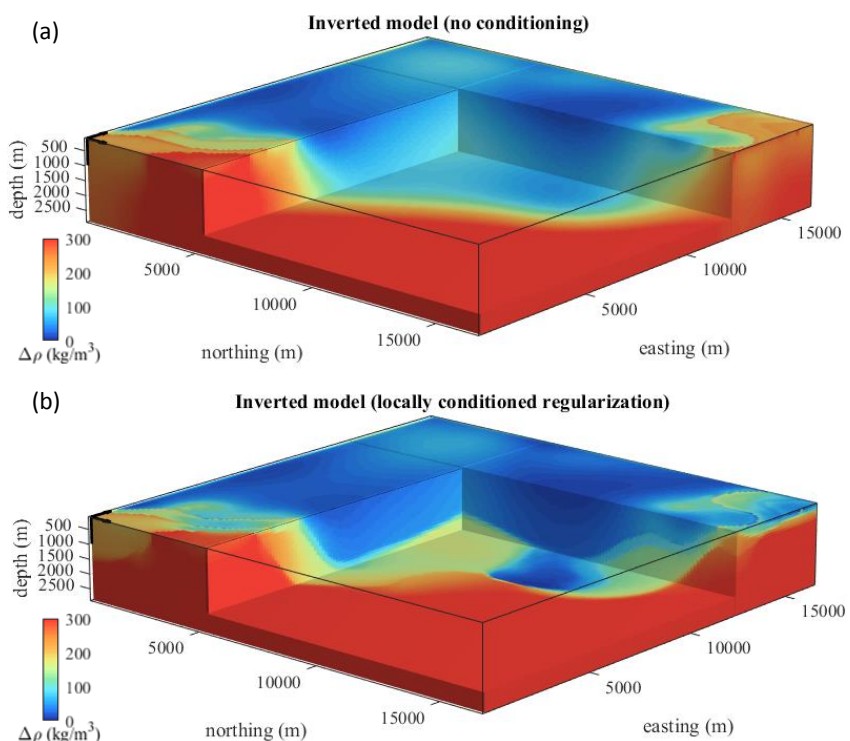

**Figure A5**. Comparison of inversion results. (a) Inverted models with non-conditioned regularization weights, and (b) using local conditioning.

Besides qualitative visual comparison of the models, we interpret inversion results through the commonly used model and data
10 root-mean-square error (RMSE). We evaluate the geometrical similarity between insverted and true model through the Bravais-Pearson correlation (also often called 'linear correlation coefficient') between their gradients (Table A1).

Comparison of the reference model (Fig. A4a) with inversion results in Fig. A5a and Fig. A5b shows that while the structures in shallowest part of the model are well retrieved in both cases, it appears that they are considerably better recovered through usage of conditioned regularization overall (Fig. A5b). The guiding effect of $W_H$ is visible in Fig. A5b where the main
15 structures at depth follow the genesral features of the conditioning volume (Fig. A4b). Moreover, in order to minimize the conditioned regularization constraint simultaneously to data misfit, inversion was driven to accommodate inverted model values (Fig. A5b) such that they are closer to the causative model (Fig. A4a) than without conditioning (Fig. A5a). This leads





to reduced model RMSE on one hand, and data RMSE on the other hand (Table A1). This reduction in data RMSE can also be explained by the relaxation of the constraints in several portions of the model, thus increasing the degree of liberty to accommodate the model towards lower data misfit. Importantly, the Bravais-Pearson correlation between the inverted and reference model gradients is nearly three times higher when information from information entropy is used, which indicates

that local conditioning of the regularization function also allows for significantly better retrieval of the causative bodies' (e.g., true model) structural features.

From these observations, we conclude that the application of the local conditioning scheme can fulfill the objectives of data integration in inversion as it is capable to recover models that are closer to the causative bodies and easier to interpret, while potentially providing reduced data misfit.

**A3 Data misfit maps from inversion in the Yerrida Basin**

Figure A6 below relates to the analysis of data misfit in Sect. 3 of the article through the plot of the data misfit maps for the non-conditioned and conditioned cases (Fig. A6d and Fig. A6h, respectively). It is complemented by the corresponding plots of starting (Fig. A6a and Fig. A6e), observed (Fig. A6b and Fig. A6f), and calculated data (Fig. A6c and Fig. A6h). Note that Fig. A6c and Fig. A6g show little visual differences, and that Fig. A6d and Fig. A6h exhibit similar features while showing

limited coherent signal.

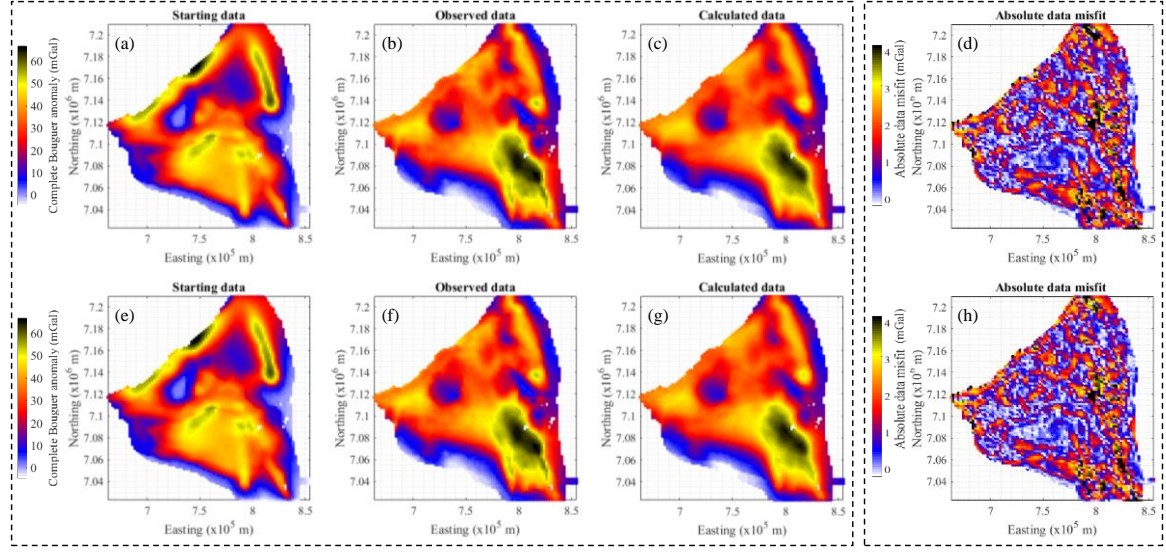

**Figure A6.** Comparison of input and output geophysical data. (a) and (e) show data calculated from the starting model, (b) and (f) input measurements, (c) and (g) data calculated from the inverted model, and (d) and (f) the absolute value of the difference of the misfit between

the observed and calculated data. (a)-(d) (e.g., first line) and (e)-(h) (e.g., second line) correspond to the non-conditioned and conditioned cases, respectively.





*Authors contribution.* Jeremie Giraud performed the integrated inverse modelling of geophysical data for both the Mansfield synthetic study and the Yerrida Basin. He performed posterior analysis and interpretation of results and he is the main contributor to the writing of this article. Mark Lindsay acquired part of the geological field measurements from the Yerrida Basin and performed the geological modelling of the area. He participated in the writing of the geological setting subsection

and he produced the geological map shown in Fig. 1a. Vitaliy Ogarko and Jeremie Giraud worked together on the implementation and testing of the proposed methodology in Tomofast-x, of which Vitaliy Ogarko, Roland Martin and Jeremie Giraud are the main developers. Mark Jessell has been involved in the validation of the methodology at the initial development stage and supervised the progress of the presented work. Roland Martin provided support at the initial stage of the inversion of gravity data from the Yerrida Basin. Evren Pakyuz-Charrier assisted Mark Lindsay with the utilisation

of MCUE. All co-authors contributed to the final version of this article. Mark Lindsay and Vitaliy Ogarko where the most actively involved in the revision process of the drafts leading to this paper.

*Competing interests*. The authors declare that they have no conflict of interest.

*Acknowledgements*. Appreciation is expressed to the CALMIP supercomputing centre (Toulouse, France), for their support through projects #P1138_2017 and #P1138_2018 and for the computing time provided on the EOS machine. Jeremie Giraud

is a recipient of the International Postgraduate Research Scholarship from the Australian Federal Government and he received a grant from the Australian Society of Exploration Geophysics Research Foundation. The authors also thank Peter Lelievre for interesting discussions and constructive feedback relating to the utilisation of gradient-related constraints. Mark D. Lindsay and Mark W. Jessell thank the Geological Survey of Western Australia (Royalties for Regions Exploration Incentive Scheme) and the Minerals Research Institute of Western Australia for their support. Mark W. Jessell is supported by a Western

Australian Fellowship. The authors acknowledge use the Zenodo research date repository to share the data necessary to reproduce the presented work.

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

10  **Table A1.** Indicators for comparison of inversion results in terms of model, data, and structure.

| | Model RMSE (kg.m⁻³) | Data RMSE (m.s⁻²) | Bravais-Pearson correlation between gradients |
|---|---|---|---|
| Non-conditioned regularization | 74.66 | $2.38*10^{-9}$ | 0.18 |
| Locally conditioned regularization | 53.05 | $7.44*10^{-10}$ | 0.53 |