# Peer review of "Integration of geological uncertainty into geophysical inversion by means of local gradient regularization"

_Solid Earth, 2018_

## Referee Comment (RC1) · C. Farquharson (Referee) · 22 Sep 2018

Dear Jérémie & Co-Authors,

I think the method that you present in this manuscript will be useful, and is correct. However, I'm not sure I agree with the reasoning behind the method and the justification you give for the method. More on this below.

But first, this is a well-written, well-organized (except for one structural issue that I'll mention later) and well-illustrated manuscript. The particular combination of geological modelling and weighted gradient term in a minimum-structure gravity/magnetic inver-

[Figure]

sion is novel, as far as I'm aware.

Okay, my thoughts on the justification of the method and the explanation of why it's working ... I really like the synthetic example given in Appendix A, especially the illustrations in Figures A4 & A5. Putting aside for a moment how W_H (shown in Figure A4b) was generated, it is exactly the case that if one weights the gradient (roughness) term in a minimum-structure term with the spatially varying weights shown in Figure A4b, then the gravity (or magnetic) inversion will construct a model for which the gradient is concentrated in the locations where W_H is small. The gravity (or magnetic) inversion is sufficiently non-unique that the data are quite happy for the gradients in the model to be put where W_H is small: the data will essentially never have a strong enough influence to overcome this effect of W_H.

This weighting of the gradient term is a bit like the weighting (well, iterative re-weighting) of the gradient term in an IRLS approach to minimize an L1 measure of roughness rather than an L2 measure of roughness. And the whole point of using an L1 measure of roughness is to get sharp interfaces between mostly uniform regions. The method you present here is kind of like asking for a sharp, L1-type interface (gradient of the model), and that this interface is located where W_H is small, i.e., you're specifying where you want this sharp interface.

I don't have any problem with this process as such. However, I'm uneasy with the connection between the regions of low W_H and your quantification of geological uncertainty. Okay, the process you create, which uses geological uncertainty to locate the low values of W_H, works. But this is because these areas of geological uncertainty (happen to?) correspond to where the boundaries between the geological units are: it's not the geological uncertainty that's the true, fundamental piece of information, it's that this uncertainty in the geological modelling is indicative of a boundary between units, and it's this estimated location of the boundary between units that becomes the key information to provide to the gravity (or magnetic) inversion via the low values of W_H.

What if you were to consider a synthetic example in which there is essentially zero uncertainty in the location of the interfaces. And make a W_H that's pretty much 1 everywhere except zero for the cells straddling the interfaces. I'd expect the gravity inversion would give a nice density model that pretty much has sharp interfaces right where the geological model has it's interfaces. If you then broaden up the zones of low values in W_H, I'd expect the boundaries in the density model to pretty much stay in the same location but now start to be smeared out and more like an inversion result for constant W_H. If you have a broad region of constant low W_H, the constructed density will be smeared out and smoothly varying through here, it won'd be sharp at one end or the other. And if you have the same true synthetic model but try putting the low values of W_H in the incorrect locations (i.e., not where the interfaces in the true model are) then the constructed density model is going to have it's interfaces (sharp or diffuse depending on whether W_H is sharp or diffuse) pretty much where the lows in W_H are, not where the boundaries are in the true model. (Have you tried such a suite of examples?)

So, yes, using spatially variable weights in the roughness term results in the interfaces in the density (or susceptibility) models occurring where you'd like them to occur. However, I don't agree with the thoughts that the gravity (magnetic) inversion is helping out, or overcoming, the geological uncertainty; rather, the uncertainty is mapping out parts of the subsurface on and close to the interfaces, but that it's simply this (fuzzy) location of the interfaces that you're using to tell the gravity (or magnetic) inversion where to put the boundaries in the density (or susceptibility) model.

The above is my main issue with this manuscript: the justification and motivation, not the mechanics of the workflow itself.

Some further comments ...

I think there has to be a typo in equation 3: You've got "max H - max H" on the denominator, i.e., a difference between identical things.

In equation 1, what are you using the prior model for? This isn't a "starting" model, is it? (You say the starting and prior are the same thing on page 7, and use "starting model", without a mention of the prior model, in the caption to Figure 2.) The m_p in equation 1 is important, as the inversion is going to try to construct a model that is close as possible to m_p (which is the whole point of that second term on the right-hand side of equation 1). This is a linear inverse problem, so it won't matter at all if one uses a starting model and then solves for the model update (or just solves directly for the model from the observed data). So a "starting" model should have no influence in the inversion. I'd definitely not use the term "starting" at all, and be careful to always use "prior" when thinking about the m_p in equation 1.

Do you now use a trade-off parameter for the "model term" in equation 1? Maybe you do in the code but it's simply been omitted from this equation when writing this manuscript?

Figure 2 and Figures 3 a & b are fine to show the whole model. But it's hard to make out the details in the parts of the model around features A, B & C. It's therefore hard to assess how much of a difference the "locally conditioned regularization" has made. I think it would be good and important to also show zoomed-in sections through the parts of the model around A, B & C.

Finally, the structural comment: I really like the example in Appendix A. I think that should come in the body of the manuscript, between sections 2 & 3 (perhaps with a description of the geological modelling process, and the process of determining the "geological uncertainty").

Best wishes, Colin Farquharson.

---

## Referee Comment (RC2) · Anonymous Referee #2 · 8 Oct 2018

Dear authors,

I think your paper is generally suitable for Solid Earth, and it is already very well written. However, there are several points where the ms needs to be improved. In particular the title and most of the rest of the ms seems to indicate that you add geological information only in the uncertainty guided inversion. Clearly, this would give additional information only for the surface structures. However, in the field example, you have used information from various geoscientific disciplines, which also add information at depth. This should be corrected throughout the ms. Furthermore, the synthetic example in the appendix is too difficult to understand with the limited information given.

[Figure]

You find more detailed comments below!

Best wishes,
Anonymous Reviewer

Specific comments:

- Section 2.2: Geological models have natural limits. Unless boreholes are available, geological observations are limited to mapping at the surface. Even though dip angles of layer interfaces measured at the surface may lead to assumptions about the depth of the interface at a given lateral offset, there is pretty poor control on this. The layer interface may not have linear depth variation, but be undulating. I recommend a general discussion of the shortcomings of geological models in terms of their uncertainties at depth.

- The synthetic example in Appendix A1 raises a number of questions and does not seem to work along the lines reported earlier on in the ms. Is the reference model in Fig. 4a your true model? Is it also used as the prior model $m_P$ in eq. 1? I guess not but in an inversion context reference and prior models are basically the same. The reader would have assumed a synthetic gravity model and independent geological information (mostly at surface cells and not so much at depth, see above). Instead the matrix $W_H$ is derived from the reference model itself, also at depth. I agree that this is helpful in showing the basic functionality of the method, bu this does not really helpful in showing the limits of the method.

- The gravity data set is limited to the NE by a fault, meaning there may be a significant density contrast right at the border of the measurement area. A comment on possible improvements in model constraints by extending the measurement area to the NE seems advisable.

- Section 3.1, p.6, l.10: Your data set is not a geological one, but a more general geoscientific one, as it includes geophysical and spectometry data. At least, I guess that the Landsat 8 and ASTER data are spectometry data and this should be mentioned clearly. So, it becomes clear here that you put much more into the matrix $W_H$ and into the reference (prior) model than what would ever be available just from geological data. Hence, the title is directly misleading. It indicates more limited scope and applicability than what you present in the paper. Please, replace "geological uncertainty" by "geoscientific uncertainty" in the title! I think this will also generate a larger exposition for your paper

- Since you include information from various geoscientific disciplines, it would be meaningful to add a larger paragraph and figures that describe what contribution the various methods make to $W_H$.

- Fig. 2b: There is structure in the $W_H$ matrix in volumes where the density contrast is zero, e.g. in the SW corner of the model and $7 * 10^5$ m E and $7.12 * 10^5$ m W (small green blob). Please explain where these anomalies in $W_H$ come from.

- Conclusions, p. 13, l.5-10: Please add a careful discussion as to whether the fact that you see predominantly the shallower part of structures A and C is a result of too little depth weighting in the inversion (e.g. Li and Oldenburg, 1996; Kamm et al., 2015).

Technical corrections:

- p.3, l.1: "In the methodology section,"

- eq. 1: Wouldn't you usually want to have another scalar factor on the model term to test different weighting of the various terms in eq. 1? Also, I wonder whether the model would not be very rough, if a diagonal $W_m$ was used and $W_H$ was set to zero in large parts of the mesh.

- eq. 2: Please provide more reasoning for this equation, in particular the log transform.

- eq. 3: Please correct this equation. As it is reproduced now, the denominator would always be zero and the argumentation in the paragraph containing eq. 3 cannot be understood.

- Fig. 1: Labels A, B and C as well as the dashed lies are supposed to indicated three greenstone belts. However, there are only two dashed lines, leading to confusion on where A and B are. This needs to be corrected.

- p.5, l.27: Please, remove "Inverted".

- p.7, l.2: "the PGM is displayed".

- p.9, l.:19-21: Kalscheuer et al. (2015) would be another suitable reference.

- p.9, l.30: "the two compared cases"

- Fig. 3: Consider replacing $\delta\|\Delta m\|^2$ by $\|\delta m\|^2$

- p.12, l.19: "The interpretation of the inversion results also reveals that greenstone B"

- p.12, l.27: "mitigates the non-uniqueness of the inversion"

- p.13, l.31: "proof of concept"

- p.14, l.13: "is shown in"

- Please define the various RMSEs based on eq. 1.

- p.15, l.15: "the general features"

- p.16, l.20: "(i.e., first line) and (e)-(h) (i.e., second line)"

**References**

T. Kalscheuer, S. Blake, J. E. Podgorski, F. Wagner, A. G. Green, M. Muller, A. G. Jones, H. Maurer, O. Ntibinyane, and G. Tshoso. Joint inversions of three types of electromagnetic data explicitly constrained by seismic observations: results from the central Okavango Delta, Botswana. *Geophys. J. Int.*, 202(3):1429–1452, 2015.

J. Kamm, I. A. Lundin, M. Bastani, M. Sadeghi, and L. B. Pedersen. Joint inversion of gravity, magnetic, and petrophysical data – A case study from a gabbro intrusion in Boden, Sweden. *Geophysics*, 80(5):B131–B152, 2015.

Y. G. Li and D. W. Oldenburg. 3-D inversion of magnetic data. *Geophysics*, 61(2):394–408, 1996.

---

## Author Comment (AC1) · 5 Nov 2018

Dear Colin,

Thank you for your insightful comments and review. We have implemented the changes in the manuscript that you requested and adjusted the text when we thought explanations could be made clearer in the light of your comments.

In particular, we have included 2 additional inversion runs in the synthetic case studies. These use uncertainty volumes where some interfaces suggested by $W_H$ are in disagreement with geology and where uncertainty is exaggerated.

We think that we have answered your concern about the justification of the integration approach we follow.

In the following, we answer to your comments relating to the methodology and to your suggestions.

Best regards,

Jérémie Giraud and co-authors

Red: review and comments.
Blue: authors' answer.
Green: text modifications.

**Comment, paragraph 1,p. C2**

- It is exactly the case that if one weights the gradient (roughness) term in a minimum-structure term with the spatially varying weights shown in Figure A4b, then the gravity (or magnetic) inversion will construct a model for which the gradient is concentrated in the locations where $W_H$ is small. The gravity (or magnetic) inversion is sufficiently non-unique that the data are quite happy for the gradients in the model to be put where $W_H$ is small: the data will essentially never have a strong enough influence to overcome this effect of $W_H$.

**Answer**

This is not exactly true. This is case-dependent in that you can well imagine a very complex geological $W_H$ model with short wavelength variations of $W_H$ where of course gravity (magnetic) inversion will not be able to overcome the effect of $W_H$. In contrast, as long as variations in $W_H$ are of similar or larger wavelength as what gravity (magnetic) data can resolve, then geophysical inversion may overcome the effect of $W_H$. This is illustrated in Fig. 3 and 4 (added to manuscript), which we remind below:

[Figure]

*Figure 1. (a) true density contrast model with outline of the 'ghost' unit B (black dashed line), embedded in ultramafic unit A, and (b) local weights calculated from the PGM calculated using MCUE for model (a).*

[Figure]

*Figure 2. Comparison of inverted model using $W_H$ derived from a PGM considering the ghost unit (b) and without it (a); (a) is the inverse model obtained without bias in the PGM as per **Error! Reference source not found.** and is shown here for comparison with (b).*

**Comment, last part paragraph 2, p. C2**

- [...] you're specifying where you want this sharp interface.

**Answer**

This remark is true to some extent but needs to be amended.

The model covariance $W_m$ is the same everywhere. This means that we do not encourage specific cells to vary independently from others. Rather, we reduce the influence between adjacent volumes of cells in the calculation of the gradient constraint by decreasing $W_H$. This allows differences between neighbouring groups of cells to be stronger than otherwise by encouraging cells in between these volumes (e.g., around interfaces or poorly constrained lithologies) to have stronger gradients. These gradients make up the interfaces. This does not mean that we are specifying where interfaces should occur, but that we specify where they are more likely to occur or where the geological information suggests they could be. This distinction is important in that when geophysical data do not support the presence of contrasts strong enough to constitute an interface, geophysical inversion does not allow this interface, and it is not observed in the inverted model. We have tested this, as illustrated in Figure 3 and 4 where we basically inserted a fictitious interface by splitting a lithology that is subsequently treated as 2 separate lithologies in the MCUE process. We then let inversion decide whether it creates an interface or anomalous body in the inverted model around the uncertain area generated by the corresponding interfaces. The result is quite clear and supports this claim. It needs, however, to be put into the context of the data inverted for. In our case, for gravity data, vertical interfaces may be well discriminated but it is likely not to be the case for mostly horizontal interfaces where it is known to show poor sensitivity. For such horizontal structures, we expect that model update will basically just follow $W_H$ and to accommodate geophysical data as you mentioned.

**Comment, paragraphs 3, p. C2**

- I don't have any problem with this process as such. However, I'm uneasy with the connection between the regions of low W_H and your quantification of geological uncertainty. Okay, the process you create, which uses geological uncertainty to locate the low values of W_H, works. But this is because these areas of geological uncertainty (happen to?) correspond to where the boundaries between the geological units are: it's not the geological uncertainty that's the true, fundamental piece of information, it's that this uncertainty in the geological modelling is indicative of a boundary between units, and it's this estimated location of the boundary between units that becomes the key information to provide to the gravity (or magnetic) inversion via the low values of W_H.

**Answer**

This is true, but to be more accurate, in general, low values of $W_H$ indicate areas of the model that are poorly constrained from a geological point of view. This comprises interfaces between 2 lithologies but also triple points and areas insufficiently 'illuminated' by geological data. In general, this is reflective of how well geology can resolve specific parts of the model, which corresponds mostly to interfaces, but can comprise whole geological units.

Added to section 2.3:

"Fundamentally, geological uncertainty contained in $H$ encapsulates information about possible locations of interfaces between units and areas where the geological data is insufficiently informative."

**Comment, paragraphs 1, p. C3**

- What if you were to consider a synthetic example in which there is essentially zero uncertainty in the location of the interfaces. And make a W_H that's pretty much 1 everywhere except zero for the cells straddling the interfaces. I'd expect the gravity inversion would give a nice density model that pretty much has sharp interfaces right where the geological model has it's interfaces. If you then broaden up the zones of low values in W_H, I'd expect the boundaries in the density model to pretty much stay in the same location but now start to be smeared out and more like an inversion result for constant W_H. If you have a broad region of constant low W_H, the constructed density will be smeared out and smoothly varying through here, it won'd be sharp at one end or the other.

**Answer**

I agree that there will be a transition between well-constrained, sharp model and the smeared model where $W_H = 1$ everywhere (e.g., no geophysical information in the constraints). We expect that there will not be a linear transition from sharp to smooth models when varying geological input uncertainty. We would expect results similar to Giraud et al. 2018, who use a model relatively similar to this one and vary uncertainty from uninformed geological models to very well constrained geology. In their case, the 'smearing' of the inverted model happens at high uncertainty (much higher than the levels tested). This work shows a non-linear relationship between the level of 'smearing' and geological uncertainty. The inversion approach they use is different (local petrophysical constraints) but the observations can be extrapolated, to a certain extent, to the present case.

For the presented case, we expect that the uncertainty level where transition between well-constrained inversion and non-constrained inversion happens corresponds to very high geological input measurement uncertainty in MCUE that are beyond what can realistically be expected. To assess this, we ran a simulation of a case where k=6, which corresponds to very uncertain data. The result is shown in Figure 5 (reproduced below).

**Comment**

- If you have a broad region of constant low W_H, the constructed density will be smeared out and smoothly varying through here, it won'd be sharp at one end or the other.

**Answer**

Figure 5 above answers partly to this remark. It holds true for the parts of the model where structures are mostly horizontal, for which gravity inversion shows little sensitivity, but not where the causative density model is shows steep structures.

Generally speaking, this observation is dependent on the sensitivity of data. For instance, replacing gravity by seismic, the opposite would be true: vertical parts would be strongly influenced by the constraints while horizontal structures would be well resolved simply because, as a rule of thumb, seismic resolves horizontal structures better than gravity.

The following has been added to the conclusion, renamed 'concluding remarks'.

"The quantitative integration technique we presented reduces uncertainty and ambiguity compared to qualitative interpretation technique or non-integrated workflows. However, despite its robustness to misplaced interface (e.g., bias) or to high geological uncertainty (e.g., sparse or very uncertain geological input measurements) as shown in the synthetic case, interpreters need to bear in mind the specificities of the geophysical data inverted for (resolution of specific geometries, depth of investigation) and the shortcomings of geological modelling workflows. As for all geological modelling, MCUE is oblivious to geological units or faults that are not sampled by field geological measurements, which can lead to biases in final models due to, for instance, inclusions not be accounted for."

**Comment**

- And if you have the same true synthetic model but try putting the low values of W_H in the incorrect locations (i.e., not where the interfaces in the true model are) then the constructed density model is going to have it's interfaces (sharp or diffuse depending on whether W_H is sharp or diffuse) pretty much where the lows in W_H are, not where the boundaries are in the true model. (Have you tried such a suite of examples?)

As hinted above, the horizontal part would be impacted and what you describe would occur. On the contrary, the vertical structures will be mostly preserved since sensitivity of gravity data is maximal for vertical anomalies. As stated earlier, this depends on the sensitivity of the method and geometry of the model. From the results we obtained, we think that, as a rule of thumb, it is better to have too many possible interfaces than too few.

We ran a series of tests, two of which are now in the manuscript (section presenting the synthetic case study); as you recommended, we move the synthetic case study directly in the text.

**Comment, paragraphs 2, p. C3:**

- So, yes, using spatially variable weights in the roughness term results in the interfaces in the density (or susceptibility) models occurring where you'd like them to occur. However, I don't agree with the thoughts that the gravity (magnetic) inversion is helping out, or overcoming, the geological uncertainty; rather, the uncertainty is mapping out parts of the subsurface on and close to the interfaces, but that it's simply this

(fuzzy) location of the interfaces that you're using to tell the gravity (or magnetic) inversion where to put the boundaries in the density (or susceptibility) model.

**Answer**

As discussed above, we have run the inversion using a $W_H$ that proposes interfaces in places where there are none in the true density contrast model.

From a more philosophical point of view, inversion is not 'correcting' geological uncertainty; rather, it accommodates and exploits it to produce an inverted model accounting for it while honouring geophysical data.

**Comment**

- The above is my main issue with this manuscript: the justification and motivation, not the mechanics of the workflow itself. Some further comments ... I think there has to be a typo in equation 3: You've got "max H - max H" on the denominator, i.e., a difference between identical things.

**Answer**

Absolutely. This has been corrected.

**Comment**

- In equation 1, what are you using the prior model for? This isn't a "starting" model, is it? (You say the starting and prior are the same thing on page 7, and use "starting model", without a mention of the prior model, in the caption to Figure 2.) The $m_p$ in equation 1 is important, as the inversion is going to try to construct a model that is close as possible to $m_p$ (which is the whole point of that second term on the right-hand side of equation 1). This is a linear inverse problem, so it won't matter at all if one uses a starting model and then solves for the model update (or just solves directly for the model from the observed data). So a "starting" model should have no influence in the inversion. I'd definitely not use the term "starting" at all, and be careful to always use "prior" when thinking about the $m_p$ in equation 1.

Thank you for pointing this out. We followed your suggestion and have addressed this issue. The equation was modified accordingly and added a short description of the term. We have also changed the semantics a little bit to avoid similar confusions around 'starting', 'reference' and 'prior' models. We made it clearer and ensured consistency with the rest of the text.

**Comment**

- Do you now use a trade-off parameter for the "model term" in equation 1? Maybe you do in the code but it's simply been omitted from this equation when writing this manuscript?

**Answer**

We do use a trade-off parameter. The way we wrote equation implicitly suggests that the trade-off parameter for the model term is included in the $W_m$ term multiplying the values initially assigned for each cell in the model covariance matrix. For clarity we rewrote equation (1) by separating the term of which the norm is calculated to stick to more conventional formulation of the cost function; the scalar weight $\alpha_m$ was introduced.

**Comment**

- Figure 2 and Figures 3 a & b are fine to show the whole model. But it's hard to make out the details in the parts of the model around features A, B & C. It's therefore hard to assess how much of a difference the "locally conditioned regularization" has made. I think it would be good and important to also show zoomed-in sections through the parts of the model around A, B & C.

**Answer**

We have tried this but adding a zoomed-section of model parts around A, B and C requires its own figure since there is not enough space left on the figure as it is. The alternative we propose is the one in the new version of the manuscript, where we added a top view of greentone belts A, B, and C to the right of the full volume view. This is helpful in providing the reader with additional information about how locally constrained regularization constrain the model and change in areas around the assumed greenstone belts.

**Comment**

- Finally, the structural comment: I really like the example in Appendix A. I think that should come in the body of the manuscript, between sections 2 & 3 (perhaps with a description of the geological modelling process, and the process of determining the "geological uncertainty").

**Answer**

We have moved it to a new section before the field case study application. In this section we also included additional examples. We have also extended this section to show some of the inversions from the suite of runs you mentioned and asked us about. This is now section 3, which has 4 subsections.

**References**

Giraud Jeremie, Pakyuz-Charrier Evren, Ogarko Vitaliy, Jessell Mark, Lindsay Mark, Martin Roland (2018) Impact of uncertain geology in constrained geophysical inversion. ASEG Extended Abstracts 2018, 1-6, https://doi.org/10.1071/ASEG2018abM1_2F

---

## Author Comment (AC2) · 5 Nov 2018

Dear Colin,

This note pertains to the revised manuscript, which you can find using the link below.

Best regards,

Jeremie

Please also note the supplement to this comment:
https://www.solid-earth-discuss.net/se-2018-79/se-2018-79-AC2-supplement.pdf

---

## Author Comment (AC3) · 5 Nov 2018

Dear Reviewer,

Thank you for your review and comments. We have applied most of your recommendations and revised the manuscript accordingly. We provide point by point answers to issues you raised in the text that follows this introductory note.

5     The main modification of the manuscript pertains to the synthetic case study. We think that the issue you raised about it came from somewhat confusing semantics in that 'reference' here referred to the reference model perturbed by MCUE, not what is commonly referred to as 'reference' model in inversion studies. We clarified this. We also added 2 examples to the synthetic case study. The first one uses a PGM that has very high uncertainty and the second one includes a fictitious lithology invisible from a petrophysical or geological point of view. We

10     think that it makes the proof of concept more meaningful and illustrates well the capabilities of the workflow.

We also made the explanation of MCUE and geological modelling clearer by providing additional more information to the previous version of the manuscript. We have extended the conclusion section, which we renamed 'concluding remarks' to add paragraphs relating to potential shortcomings of our geological modelling approach and discussing depth weighting.

15     Best regards,

Jérémie Giraud and co-authors.

Red: reviewer's comment.

Blue: author's answer

20     Green: modifications in the text.

**Comment**

Dear authors,

I think your paper is generally suitable for Solid Earth, and it is already very well written. However, there are

25     several points where the ms needs to be improved. In particular the title and most of the rest of the ms seems to indicate that you add geological information only in the uncertainty guided inversion. Clearly, this would give additional information only for the surface structures.

**Answer**

Your statement about geological information being provided only for the surface structures needs to be amended.

30     More specifically, our geological modelling scheme provides information about the structures that are accessible from surface (or borehole). Geological data and uncertainty are propagated downwards to calculate geological models by MCUE. Therefore, we infer the parts of the model that are not accessible from surface of borehole and their related uncertainty. This process is non-linear, resulting in uncertainty models showing features much more

complex that a fuzzification of the model used as reference for MCUE (this process is detailed in the references provided).

It seems that more background information need to be given to readers curious about this method. We modified the first sentence of subsection 2.2 as:

"The sampling algorithm perturbs orientation data used to derive a reference model by sampling probability distributions describing the uncertainty of orientation data to produce a series of unique altered geological models", and:

"Probabilistic geological modelling is performed using the Monte-Carlo Uncertainty Estimator (MCUE) method of (Pakyuz-Charrier et al., 2018; Pakyuz-Charrier et al., 2018), which extends previous works from (Jessell et al, 2010; Lindsay et al., 2012; Wellmann et al., 2010)."

The additional references provided also add justification for the validity of the method used.

To make the description more accurate, we have added the following to our introduction of the methodology in subsection 2,2: "foliation and interface of the geological units sampled at surface level or in borehole".

We also added to this subsection: "Coupled to the 3D geological modelling engine of Geomodeller© (Calcagno et al, 2008), it produces a set of plausible geological models honouring the geological input measurements that representing the geological model space (Lindsay et al., 2013)".

The conclusion section has been modified to account with this comment (see answer to comment relating to the conclusion below).

**Comment**

- However, in the field example, you have used information from various geoscientific disciplines, which also add information at depth.

**Answer**

We have indeed used several datatypes to build the geological reference model (e.g., unperturbed) from which the geological input measurements where subsequently perturbed by MCUE. The different datasets used to build this geological model are clearly stated in section 4.1. In the light of your comment, we added information about the utilisation of the different datasets used to derive the reference geological model. This has been addressed in the comments below (marked with *ans*)

**Comment**

- This should be corrected throughout the ms. Furthermore, the synthetic example in the appendix is too difficult to understand with the limited information given.

**Answer**

There was possible confusion stemming from the lexicon used in the appendix, e.g, 'reference', 'starting', 'prior' models. We made it clearer and ensured consistency with the rest of the text. We also moved this appendix as a separate section in the text.

We modified the section describing the synthetic model and added 2 subsections to it. They show additional tests that we think answer some of your later comments. It is not an appendix anymore. It is part of the text as section 3. We refer you directly to section 3 for the modifications brought to the text.

**Comment**

Specific comments:

- Section 2.2: Geological models have natural limits. Unless boreholes are available, geological observations are limited to mapping at the surface. Even though dip angles of layer interfaces measured at the surface may lead to assumptions about the depth of the interface at a given lateral offset, there is pretty poor control on this. The layer interface may not have linear depth variation, but be undulating. I recommend a general discussion of the shortcomings of geological models in terms of their uncertainties at depth

**Answer**

We have added a paragraph in the conclusion section to the manuscript, where we discuss the limitations of our methodology in terms of geological modelling. We renamed the conclusion section 'concluding remarks'.

The following was added:

"The quantitative integration technique we presented reduces uncertainty and ambiguity compared to qualitative interpretation technique or non-integrated workflows. However, despite its robustness to misplaced interface (e.g., bias) or to high geological uncertainty (e.g., sparse or very uncertain geological input measurements) as shown in the synthetic case, interpreters need to bear in mind the specificities of the geophysical data inverted for (resolution of specific geometries, depth of investigation) and the shortcomings of geological modelling workflows. As for all geological modelling, MCUE is oblivious to geological units or faults that are not sampled by field geological measurements, which can lead to biases in final models due to, for instance, inclusions not be accounted for.

Current research comprises the development of sensitivity and resolution analyses in an effort to mitigate the risk of the model being affected by unaccounted for uncertainty sources."

**Comment**

- The synthetic example in Appendix A1 raises a number of questions and does not seem to work along the lines reported earlier on in the ms. Is the reference model in Fig. 4a your true model? Is it also used as the prior model $m_P$ in eq. 1? I guess not but in an inversion context reference and prior models are basically the same. The reader would have assumed a synthetic gravity model and independent geological information (mostly at surface cells and not so much at depth, see above). Instead the matrix WH is

derived from the reference model itself, also at depth. I agree that this is helpful in showing the basic functionality of the method, bu this does not really helpful in showing the limits of the method.

**Answer**

Yes, Fig. 4a is the true model. It is not used as prior model. The prior model is set to 0 kg/m3. Quoting the manuscript "Please note that, simulating the absence of prior petrophysical information, a homogenous starting model set to 0 kg.m$^{-3}$ was used in both cases."

To avoid such confusion, we changed 'reference' for 'true' model when it comes to density contrast models, and maintained the term 'reference' only for the reference geological that is perturbed my MCUE.

$W_h$ is derived from the PGM, which is calculated through MCUE applied to the reference model and data used to derive it.

The limits of the method have been investigated and 2 different case scenarios have been added to the synthetic case:

1) with an interface where there is no density contrast (Fig. 3 and 4)

2) with exaggerated geological data input uncertainty (Fig. 5).

It would be possible to perform a complete sensitivity analysis of the method to uncertainty in geological input data and biases but this is not the object of this article and we believe that such work deserves a separate publication.

**Comment**

- The gravity data set is limited to the NE by a fault, meaning there may be a significant density contrast right at the border of the measurement area. A comment on possible improvements in model constraints by extending the measurement area to the NE seems advisable.

**Answer**

It is true that additional constraints can be gained by expanding the boundaries of the current model. The newly expanded model can be analysed, and boundaries expanded again to accommodate some other unexpected misfit that is potentially solvable with again expanded boundaries, but doesn't answer the original question any better than the original boundary parameters. In addition, this particular boundary was set because the geology on the other side of the NE fault is the Bryah Basin, which has undergone a different deformation history than the Yerrida Basin. Deformation was much more intense and produced folding and geometries that are very difficult to model with implicit methods and Geomodeller. Attempt to reproduce this geometry failed, and, given the target was the Yerrida Basin anyway, the model was restricted to the current bounds.

**Comment**

- Section 3.1, p.6, l.10: Your data set is not a geological one, but a more general geoscientific one, as it includes geophysical and spectometry data. At least, I guess that the Landsat 8 and ASTER data are spectometry data and this should be mentioned clearly. So, it becomes clear here that you put much more

into the matrix WH and into the reference (prior) model than what would ever be available just from geological data. Hence, the title is directly misleading. It indicates more limited scope and applicability than what you present in the paper. Please, replace "geological uncertainty" by "geoscientific uncertainty" in the title! I think this will also generate a larger exposition for your paper.

**Answer**

We applied this modification.

Remote sensing data was used for quality control of consistency of surface interpretation of lithologies. To avoid confusion of the reader, we have added the following to the description of the geological context: "…landsat 8 and ASTER hyperspectral data".

**Comment**

- Since you include information from various geoscientific disciplines, it would be meaningful to add a larger paragraph and figures that describe what contribution the various methods make to WH

**Answer**

We have added information about the contribution of the different techniques to the reference geological model that was fed to MCUE. However, we restrict this information to explanation in the text as it may extend the paper more than necessary. The other disciplines were not extensively used to derive the geological model per se but rather as a QC tool and to remove regional trends present in the data (*ans*).

Primary data sources for the 3D geological model were geological maps and structural measurements, and structural interpretation of magnetic data. Gravity data was not used during interpretation. Remote-sensing data was used to identify areas where regolith was present so that any associated geophysical anomaly was not attributed to bedrock and then input to the 3D geological model. Thus remote sensing data supported the geological inputs, but was not an input itself.

The main modifications to the geological modelling section are shown below.

The following was added to section 4.2: "Remote-sensing data was used to confirm interpretations".

"These datasets were used jointly to build a reference geological model reconciling the available geological information in Geomodeller."

**Comment**

- Fig. 2b: There is structure in the WH matrix in volumes where the density contrast is zero, e.g. in the SW corner of the model and $7 * 10^5$ m E and $7{:}12 * 10^5$ m W (small green blob). Please explain where these anomalies in WH come from.

**Answer**

These anomalies are moderate deviations from $W_H = 1$, meaning that they present relatively low uncertainty. The anomaly in the SW corresponds to a part that is data-poor compared to the rest of the area, meaning that it is

poorly constrained, while the one at ~ $7 * 10^5$ m E and $7.12 * 10^5$ m W represents the bottom part of an uncertain region.

**Comment**

- Conclusions, p. 13, l.5-10: Please add a careful discussion as to whether the fact that you see predominantly the shallower part of structures A and C is a result of too little depth weighting in the inversion (e.g. Li and Oldenburg, 1996; Kamm et al., 2015).

**Answer**

We do not think that this is necessary. Nevertheless, we added the following at the end of section 2.1.:

"We utilize the integrated sensitivities technique of (Portniaguine & Zhdanov, 2002) to balance the decreasing sensitivity of gravity data with depth. We chose this technique because it offers the advantage of providing 'equal sensitivity of the observed data to the cells located at different depths and at different horizontal positions' (Vatankhah & Renaut, 2017)."

We removed the text related to depth weighting in section 4.1.:

""

The fact that most of the density contrasts are located close to surface comes from several factors. First, the regional trends have been removed from the data. This means that not having deep, long wavelength anomalies is not in contradiction with this fact. Second, the geological model is such that most units that may present a strong density contrast are actually located close to the surface.

**Comment**

Technical corrections

**Answer**

We provide an answer only to recommendations we have not followed or which require us to answer.

**Comment**

- eq. 1: Wouldn't you usually want to have another scalar factor on the model term to test different weighting of the various terms in eq. 1? Also, I wonder whether the model would not be very rough, if a diagonal Wm was used and WH was set to zero in large parts of the mesh.

**Answer**

Yes. We have added that scalar term in the equation.

A test with zero values for the constraining volumes in the large parts of the mesh is show below:

[Figure]

We simulated broadly exaggerated uncertainty and set to 0 in the Wh volume of values inferior to 0.05 equal to zero (black portions of the model).

**Comment**

- eq. 2: Please provide more reasoning for this equation, in particular the log transform.

**Answer**

We do not think that it is necessary. This metric is not new and has already been used by the references given that support its usage (~10 references). Shannon entropy (Shannon, 1948) is not a new concept. It was generalised by (Rényi, 1961) and has become used in a number of fields.

**Comment**

- Fig. 3: Consider replacing $\delta \|\Delta m\|^2$ by $\| \delta m\|^2$

**Answer**

We have not implemented this change to keep notation between (d) and (e) of that figure consistent.

**References:**

Calcagno, P., Chilès, J. P., Courrioux, G., & Guillen, A. (2008). Geological modelling from field data and geological knowledge. Part I. Modelling method coupling 3D potential-field interpolation and geological rules. *Physics of the Earth and Planetary Interiors*, *171*(1‑4), 147‑157. https://doi.org/10.1016/j.pepi.2008.06.013

Jessell, M. W., Ailleres, L., & de Kemp, E. A. (2010). Towards an integrated inversion of geoscientific data: What price of geology? *Tectonophysics*, *490*(3‑4), 294‑306. https://doi.org/10.1016/j.tecto.2010.05.020

Lindsay, M. D., Aillères, L., Jessell, M. W., de Kemp, E. A., & Betts, P. G. (2012). Locating and quantifying geological uncertainty in three-dimensional models: Analysis of the Gippsland Basin, southeastern

Australia. *Tectonophysics*, *546‑547*, 10‑27. https://doi.org/10.1016/j.tecto.2012.04.007

Lindsay, M. D., Jessell, M. W., Ailleres, L., Perrouty, S., de Kemp, E., & Betts, P. G. (2013). Geodiversity: Exploration of 3D geological model space. *Tectonophysics*, *594*(September 2015), 27‑37. https://doi.org/10.1016/j.tecto.2013.03.013

Pakyuz-Charrier, E., Giraud, J., Ogarko, V., Lindsay, M., & Jessell, M. (2018). Drillhole uncertainty propagation for three-dimensional geological modeling using Monte Carlo. *Tectonophysics*. https://doi.org/10.1016/j.tecto.2018.09.005

Pakyuz-Charrier, E., Lindsay, M., Ogarko, V., Giraud, J., & Jessell, M. (2018). Monte Carlo simulation for uncertainty estimation on structural data in implicit 3-D geological modeling, a guide for disturbance distribution selection and parameterization. *Solid Earth*, *9*(2), 385‑402. https://doi.org/10.5194/se-9-385-2018

Portniaguine, O., & Zhdanov, M. S. (2002). 3-D magnetic inversion with data compression and image focusing. *GEOPHYSICS*, *67*(5), 1532‑1541. https://doi.org/10.1190/1.1512749

Rényi, A. (1961). On measures of entropy and information. In *Fourth Berkeley Symposium on Mathematical Statistics and Probability*. https://doi.org/10.1021/jp106846b

Shannon, C. E. E. (1948). A Mathematical Theory of Communication. *Bell System Technical Journal*, *27*(3), 379‑423. https://doi.org/10.1002/j.1538-7305.1948.tb01338.x

Vatankhah, S., & Renaut, R. A. (2017). Comment on: 'Improving compact gravity inversion based on new weighting functions', by Mohammad Hossein Ghalehnoee, Abdolhamid Ansari and Ahmad Ghorbani. *Geophysical Journal International*, *211*(1), 346‑348. https://doi.org/10.1093/gji/ggx058

Wellmann, J. F., Horowitz, F. G., Schill, E., & Regenauer-Lieb, K. (2010). Towards incorporating uncertainty of structural data in 3D geological inversion. *Tectonophysics*, *490*(3‑4), 141‑151. https://doi.org/10.1016/j.tecto.2010.04.022

---

## Author Response (AR1)

Dear Editorial Board,

We have addressed all the comments of the reviewers in a detailed point-by-point fashion. In our answer to the reviewers, we detail the modifications we made to the manuscript and answered to each of their comments. These answers are available publicly in the answer to the reviewers, which we posted as part of the interactive discussion. We therefore do not add a copy of these answers to our response here.

The rest of this document is organised as follows. The version of the manuscript represents the initial version, where what is shown in red symbolises parts of the manuscript which we removed/replaced. The second version is the revised version where what is shown in green marks what we have added. The third version is the comparison report where insertions (green), deletions (red) and replacements (purple) are highlighted.

Please do not hesitate to ask if you need further information.

Best regards,

The Authors

[revised manuscript text omitted]
 parameters $\alpha_m$ and $\alpha_H$ control the overall weight of model and structural regularization terms; $\nabla$ is the spatial gradient operator. Note that $\|\nabla\boldsymbol{m}\|_2$, estimates the amount of structure in inverted physical property model $\boldsymbol{m}$. Also note that parts of the model where $\boldsymbol{W_H} = 0$ are excluded from the calculation of the structural regularization and can change freely to accommodate geophysical data.

We utilize the integrated sensitivities technique of (Portniaguine and Zhdanov, 2002) to balance the decreasing sensitivity of gravity data with depth. We chose this technique because it offers the advantage of providing "equal sensitivity of the observed data to the cells located at different depths and at different horizontal positions" (Vatankhah and Renaut, 2017).

**2.2 Probabilistic geological modelling**

Probabilistic geological modelling is performed using the Monte-Carlo Uncertainty Estimator (MCUE) method of (Pakyuz-Charrier et al., 2018b, 2018a), which extends previous works from (Jessell et al., 2010; Lindsay et al., 2012; Wellmann et al., 2010). It is a 3D uncertainty propagation method for implicit geological modelling that uses geological rules and geological orientation measurements (foliation and interface of the geological units sampled at surface level or in borehole) as inputs. The sampling algorithm perturbs orientation data used to derive a reference model by sampling probability distributions describing the uncertainty of orientation data to produce a series of unique altered geological models. Realizations that do not respect a series of geological rules are considered implausible and are rejected. Coupled to the 3D geological modelling engine of Geomodeller© (Calcagno et al., 2008), it produces a set of plausible geological models honouring the geological input measurements that represent the geological model space (Lindsay et al., 2013b). The observation probabilities constituting the probabilistic geological model (PGM) are obtained, in each model cell, by calculating the relative observation frequencies of the different lithologies from the set of geological models. For the $i^{\text{th}}$ model cell of a PGM containing $L$ lithologies, vector $\boldsymbol{\psi^i} = \left[\psi_{k=1}^i, \dots, \psi_{k=L}^i\right]$ contains the observation probabilities of each lithology. As we show in the next subsection, the resulting PGM can be used to estimate uncertainty levels and as a source of prior information.

**2.3 Utilisation of information entropy for local conditioning**

Information entropy was introduced for geological modelling by Wellmann and Regenauer-Lieb (2012) and is gaining popularity as a measure of uncertainty in probabilistic geological modelling (de la Varga et al., 2018; de la Varga and Wellmann, 2016; Lindsay et al., 2014; Lindsay et al., 2013; Pakyuz-Charrier et al., 2018; Schweizer et al., 2017; Thiele et al., 2016; Wellmann et al., 2017; Yamamoto et al., 2014). Quoting (Schweizer et al., 2017), information entropy "quantifies the amount of missing information and hence, the uncertainty at a discrete location". For the $i^{\text{th}}$ model-cell, it is given as (Shannon, 1948):

$$H^i = H\left(\boldsymbol{\psi^i}\right) = -\sum_{k=1}^{L} \psi_k^i \log\left(\psi_k^i\right). \tag{2}$$

Fundamentally, geological uncertainty contained in $\boldsymbol{H}$ encapsulates information about possible locations of interfaces between units and areas where the geological data is insufficiently informative. Instead of using $\boldsymbol{H}$ directly, we calculate $\boldsymbol{W_H}$ utilising

its normalized complementary, which reflects the degree of certainty across the model. Let us express $\boldsymbol{W_H}$ as follows, for the $i^{\text{th}}$ model cell:

$$W_H^i = \frac{\max \boldsymbol{H} - H^i}{\max \boldsymbol{H} - \min \boldsymbol{H}} \tag{3}$$

The consequence of Eq. (2) and 3 is that $\boldsymbol{W_H}$ is minimum at interfaces and in areas poorly constrained by geological information, and equal to unity in areas where the geology is well resolved. Consequently, the conditioning process attaches small weights to the structural term of Eq. (1) in uncertain cells, while consistently high values will be applied to the most geologically certain cells. As a result, it enables the inversion algorithm to favour nearly constant changes in the inverted model in contiguous certain groups of cells (e.g., where $\boldsymbol{W_H} \rightarrow 1$) while relaxing the regularisation constraints in the most uncertain parts (e.g., where $\boldsymbol{W_H} \rightarrow 0$).

**3    Proof of concept: synthetic case study**

This section introduces the proof of concept of the proposed method through an idealized case study illustrating the potential of the proposed inverse modelling scheme to ameliorate inversion results and to reduce interpretation uncertainty. We use the same 3D density contrast model as (Giraud et al., 2017), which is obtained by populating the structural framework of (Pakyuz-Charrier et al., 2018b). We simulate a series of PGM sought to represent expected values as well as possible extreme scenarii. The short presentation of the model below and the analysis of results provides essential information about the synthetic survey and shows the proof of concept of the methodology used in the paper.

**3.1    Survey setup**

The 3D unperturbed reference geological model was generated from contact (interface points) and surface orientation (foliations) field measurements collected in the Mansfield area (Victoria, Australia). It presents a Carboniferous mudstone-sandstone basin oriented N170, abutting a faulted contact with a folded ultramafic basement to the South-West. Model complexity was artificially increased through the addition of a North-South fault and of a mafic intrusion.

The true density contrast model (Figure 1a) was obtained assigning density contrasts consistently with the structural setting of the reference geological model, assuming a flat topography. Density contrasts of 0 and 100 kg.m$^{-3}$ were assigned to the upper and lower basin lithotypes, respectively. Mafic rocks were assigned a density contrast of 200 kg.m$^{-3}$ while the density contrast of the ultramafic basement was set to 300 kg.m$^{-3}$.

MCUE perturbations of the reference geological model were first performed using standard measurement uncertainty values recommended by metrological studies as reported by (Allmendinger et al., 2017; Novakova and Pavlis, 2017). We generated a series of 300 models that were subsequently combined into a PGM. The resulting volume representing the $\boldsymbol{W_H}$ values calculated from this PGM in each cell of the model as per Eq. (3) is shown in Figure 1b.

[Figure]

**Figure 1.** True density contrast model and $W_H$ values used for local regularization conditioning. (a) Unperturbed reference model populated with density contrast values, (b) uncertainty values used for local regularization conditioning.

**3.2    Locally constrained inversion: validation**

5   To assess the impact of local conditioning of the regularization function onto inversion, we compare inversions using non-conditioned (Figure 2a) and locally conditioned (Figure 2b) regularization function, respectively. Please note that, simulating the absence of prior petrophysical information, a homogenous prior model set to 0 kg.m-³ was used in both cases.

[Figure]

(a)

**Inverted model (no conditioning)**

(b)

**Inverted model (locally conditioned regularization)**

**Figure 2**. Comparison of inversion results. (a) Inverted models with non-conditioned regularization weights, and (b) using local conditioning.

Besides qualitative visual comparison of the models, we interpret inversion results (Figure 2) through the commonly used model and data root-mean-square error (RMSE), which correspond to the model and data terms calculated with weights and

5    covariances set to unity. We evaluate the geometrical similarity between inverted and true model through the Bravais-Pearson correlation (also often called 'linear correlation coefficient') between their gradients (Table 1).

Comparison of the true model (Figure 1a) with inversion results in Figure 2a and Figure 2b shows that while the structures in shallowest part of the model are well retrieved in both cases, it appears that they are considerably better recovered through usage of conditioned regularization overall (Figure 2b). The guiding effect of $W_H$ is visible in Figure 2b where the main

10   structures at depth follow the general features of the conditioning volume (Figure 1b). Moreover, in order to minimize the conditioned regularization constraint simultaneously to data misfit, inversion was driven to accommodate inverted model values (Figure 2b) such that they are closer to the causative model (Figure 1a) than without conditioning (Figure 2a). This leads to reduced model RMSE on the one hand, and data RMSE on the other (Table 1). This reduction in data RMSE can also be explained by the relaxation of the constraints in several portions of the model, thus increasing the degree of liberty to

15   accommodate the model towards lower data misfit. More importantly, the Bravais-Pearson correlation between the inverted and true model gradients is much higher when information from information entropy is used. This indicates that local

conditioning of the regularization function also allows for significantly better retrieval of the causative bodies' (e.g., the true model) structural features.

Please also note that we do not show the recovered geophysical measurements because visual differences between recovered and inverted measurements are minimal.

**Table 1.** Indicators for comparison of inversion results in terms of model, data, and structure.

|  | Model RMSE (kg.m⁻³) | Data RMSE (m.s⁻²) | Correlation between gradients |
|---|---|---|---|
| *Non-conditioned regularization* | 74.66 | $2.38 \times 10^{-9}$ | 0.18 |
| *Locally conditioned regularization* | 53.05 | $7.44 \times 10^{-10}$ | 0.53 |

From these observations, we conclude that the application of the local conditioning scheme can fulfil the objectives of data integration in inversion as it allows inversion to recover models that are closer to the causative bodies and easier to interpret, while honouring geophysical data. Nevertheless, it remains important to test the methodology in cases where the uncertainty indicator $W_H$ is biased and/or shows high geological uncertainty levels away from faults and contacts. A thorough analysis lying beyond the scope of this paper, the remainder of this section presents an elementary sensitivity analysis using a series of two $W_H$ volumes representing distinct extreme scenarios.

**3.3 Inversion constrained by biased geological uncertainty model**

In this subsection, we investigate the effect of inaccurate geological models and the propagation of the related uncertainty in inversion. For this purpose, we calculate a second PGM from MCUE perturbations in which we split the ultramafic basement into two independent units, without changing the density contrast values (Figure 3a). This results in the existence of a fictitious geological unit that is invisible to gravity data and presents no density contrast but which increases geological complexity and uncertainty (Figure 3b) (we further refer to it as 'ghost' geological unit). Notably, it increases geological uncertainty and smears interfaces that are well-constrained as per Figure 1. It also decreases $W_H$ in large parts of the model where $W_H \rightarrow 1$ previously, thereby favouring model changes in these areas during inversion and encouraging it to place larger density contrast or interfaces in these areas.

[Figure]

**Figure 3.** (a) true density contrast model with outline of the 'ghost' unit B (black dashed line), embedded in ultramafic unit A, and (b) local weights calculated from the PGM calculated using MCUE for model (a).

Comparison of the inverted models obtained without (Figure 4a) and with the ghost unit (Figure 4b) reveals that they exhibit broadly similar features except in the most geologically complex parts of the model as per Figure 1b, where differences are minor. This indicates that while geophysical inversion updates the prior density contrast model preferably in geologically uncertain regions, low values of $W_H$ do not necessarily lead to the modelling of an interpretable interface by inversion. From the comparison of Figure 3b and Figure 4b, we can deduce that local conditioning as applied in this work does not necessarily enforce strict structural similarity between the inverted model and the conditioning geological uncertainty volume.

[Figure]

**Figure 4.** Comparison of inverted model using $W_H$ derived from a PGM considering the ghost unit (b) and without it (a); (a) is the inverse model obtained without bias in the PGM as per Figure 2 and is shown here for comparison with (b).

**3.4    Inversion constrained using exaggerated geological uncertainty**

5    To complete this series of tests, we generated a third PGM showing exaggerated geological uncertainty. To this end, we used a half aperture 95% confidence interval of ~50 degrees for orientation data measurements in our MCUE simulations. This is far higher than for the rest of the MCUE simulations used in this paper. All other simulations (Subsect. 3.2 and 3.3) use a value of ~11 degrees, which is representative of realistic measurement uncertainty as proposed by recent metrological studies (Allmendinger et al., 2017; Cawood et al., 2017; Novakova and Pavlis, 2017). Figure 5 below shows the resulting $W_H$ volume

10    (Figure 5a) and the inverted model obtained using it for local conditioning of the regularization constraint (Figure 5b).

[Figure]

**Figure 5.** (a) Local weights $W_H$ calculated from a PGM corresponding to exaggerated uncertainty in geological input measurements and (b) corresponding inverted density contrast model.

5     The features visible in Figure 5a reflect the high geological input measurement uncertainty. Geologically uncertain areas cover large portions of the volume and only the simplest geological structures (e.g., the basin) seem to be well constrained by geology. Areas of the model previously well constrained (Figure 1b) present varying degrees of uncertainty. This illustrates that, as can be expected, increasing geological input uncertainty translates in relaxing the guiding effect of local conditioning using $W_H$, which results in geophysical inversion being less strongly guided by geological information. As can be seen in Fig

10     4b, the inverted model obtained in this case shows structures that present weaker contrast around interfaces than when geological uncertainty is lower (Figure 2b). Importantly, however, most structures are well preserved and the overall model values for the different lithologies remain closer to the true model than for the non-conditioned case (Figure 2a). This indicates that even in high geological uncertainty scenarios, interpretation outcome may be largely more reliable when local regularisation is used.

The analysis and comparison of the results shown in this section demonstrates the potential of the proposed inverse modelling scheme to ameliorate inversion results and to reduce interpretation uncertainty. It also illustrates the capability of our methodology to deal with high or biased conditioning uncertainty estimates. In this synthetic case, local conditioning allows

geophysical inversion to significantly improve the imaging of geologically uncertain areas and to exploit complementarities between geological modelling and geophysical inversion. From the success of this proof of concept study, we are confident that our integration method can be tested here using real world data for field validation.

**4    Field validation: Yerrida Basin case study**

**4.1    Geological context and geophysical survey setup**

The Yerrida Basin is located in the southern part of the Capricorn Orogen, at the northern margin of the Yilgarn Craton in Western Australia (Figure 6a), and extends approximately 150km N-S and 180 km E-W (Figure 6b). The studied area is bounded in the northwest by the Goodin Fault, which represents a faulted contact between the Yerrida Basin and the Bryah-Padbury Basin. The structures of interest in this work are the Archean greenstone belts extending north-northwest that are unconformably overlain by Paleoproterozoic sedimentary rocks the form the Yerrida Basin. Features A and B (Figure 6a and Figure 6b) indicate the interpreted position of buried Wiluna Greenstone Belt. Where the Wiluna Greenstone Belt is exposed, it is host to base and precious metal mineralisation (Williams, 2009). With a relatively high positive density contrast (expected to be between 190 and 270 kg.m⁻³) to the Yilgarn Craton granite-gneiss host, mafic greenstone belts A, B, and C are suitable targets for gravity inversion. Interpretations from multiple studies in the region, e.g, (Johnson et al., 2013; Pirajno et al., 1998; Pirajno and Adamides, 2000; Pirajno and Occhipinti, 2000) were compiled while additional geological measurements acquired in 2015, 2016 and 2017 complemented legacy data (Occhipinti et al., 2017; Olierook et al., 2018). This allowed the revision of existing models and improved our understanding of the area. This, in turn, also highlights the challenges presented by the characterization of greenstone belts A, B and C, and that further geophysical analysis through constrained inversion is a useful pathway for reducing exploration risk.

Geophysical data consists of ground surveys obtained from Geoscience Australia (http://www.ga.gov.au/data-pubs). Post-processing includes spherical-cap and terrain corrections and the removal of the regional trend to obtain the complete Bouguer anomaly. As most data points were acquired 1 to 4 km apart, the dataset was resampled to match the geological model discretization, making up a total of 4882 measurement points. The model is discretized into $100 \times 100 \times 42$ cells of dimensions 2.335 km $\times$ 1.875 km $\times$ 1.0475 km, down to a depth of 44 km, making up a total of 420000 cells.

[Figure]

**Figure 6.** Geological context and geophysical data. (a) Geological map of the area and (b) complete Bouguer anomaly. The dashed lines delineate the possible sub-basin extent of the mafic greenstone belts A, B and C.

**4.2    Geological modelling**

Geological information consists of in-situ structural measurements (interfaces and foliations) and interpretation of aeromagnetic, airborne electromagnetic, Landsat 8 and ASTER hyperspectral data. Legacy data from the Geological Survey of Western Australia (Pirajno and Adamides, 2000) and CSIRO (Ley-Cooper et al., 2017) were used, to which about 600 surface geological and petrophysical measurements from recent geological field campaigns were added. Although the available petrophysical measurements were not used to derive petrophysical constraints because of the insufficient sampling of several of the modelled lithologies, they were a useful source of information to populate geological models and for interpretation. Remote-sensing data were used to test interpretations.

These datasets were used jointly to build a reference geological model reconciling the available geological information in Geomodeller.

Lithologies with similar density contrasts were merged and subsequently treated as a single rock type in MCUE simulations. Uncertainty related to structural measurements was subsequently used as inputs to the MCUE perturbations (Pakyuz-Charrier et al., 2018b) of the reference model to generate a collection of 500 accepted models. Information extracted from the PGM is displayed in Figure 7. Figure 7a shows the lithologies with the highest probability for each cell of the PGM. The associated $W_H$ values used in inversion are shown in Figure 7b. The prior model for inversion $\boldsymbol{m_p}$ is equal to the mean model of the 500 plausible models generated by MCUE. It is shown in Figure 7c.

[Figure]

**Figure 7.** Geological modelling results. (a) Most probable lithology in each model cell (same colour code as in Figure 6) (b) values used for local regularization conditioning, (c) and prior model derived from PGM and prior petrophysical information). In (a), "background" refers to all the lithologies that have a density contrast equal to 0 kg.m⁻³.

**4.3 Inversion results and analysis**

The aim of our analysis is to assess the influence of the local conditioning of structural constraints on inversion through comparison with the non-conditioned case, all other things remaining constant.

**4.3.1 Comparative analysis strategy**

Prior to examination of the inverted models, we analyse geophysical data misfit after inversion. This enables us to ensure that the inversion results we compare produce, in our case, similar gravity anomalies. Our study of inverted models focuses on results obtained through usage of non-conditioned (Figure 8a) and conditioned regularization function (Figure 8b) using $W_H$ (Figure 7b). In addition to departures from the prior 
[revised manuscript text omitted]

The quantitative integration technique we presented reduces uncertainty and ambiguity compared to qualitative interpretation technique or single-discipline workflows. However, despite its robustness to misplaced interface (e.g., bias) or to high geological uncertainty (e.g., sparse or very uncertain geological input measurements) as shown in the synthetic case, interpreters need to bear in mind the specificities of the geophysical data inverted for (resolution of specific geometries, depth of investigation) and the shortcomings of geological modelling workflows. As for all geological modelling, MCUE is oblivious to geological units or faults that are not sampled by field geological measurements, which can lead to biases in final models due to, for instance, inclusions not be accounted for.

Current research comprises the development of sensitivity and resolution analyses in an effort to mitigate the risk of the model being affected by uncertainty sources not accounted for. Future research will include the utilization of local petrophysical constraints of (Giraud et al., 2017) in the uncertainty-guided inversion scheme we presented, as well as the utilisation of geological uncertainty to weight the cross-gradient term of (Gallardo and Meju, 2003) locally. With this last respect, uncertainty-guided inversion can be assisted in the most uncertain parts of the model by guided inversion (in the sense of Brown et al., 2012) or through cross-gradient joint inversion.

[revised manuscript text omitted]

Wellmann, J. F., de la Varga, M., Murdie, R. E., Gessner, K. and Jessell, M.: Uncertainty estimation for a geological model of the Sandstone greenstone belt, Western Australia – insights from integrated geological and geophysical inversion in a Bayesian

inference framework, Geol. Soc. London, Spec. Publ., SP453.12, doi:10.1144/SP453.12, 2017.

Wiik, T., Nordskag, J. I., Dischler, E. Ø. and Nguyen, A. K.: Inversion of inline and broadside marine controlled-source electromagnetic data with constraints derived from seismic data, Geophys. Prospect., 63(6), 1371–1382, doi:10.1111/1365-2478.12294, 2015.

Williams, N. C.: Geologically-constrained UBC-GIF gravity and magnetic inversions with examples from the Agnew-Wiluna greenstone belt, Western Australia, University of British Columbia. [online] Available from: https://open.library.ubc.ca/cIRcle/collections/ubctheses/24/items/1.0052390, 2008.

Williams, N. C.: Mass and magnetic properties for 3D geological and geophysical modelling of the southern Agnew–Wiluna Greenstone Belt and Leinster nickel deposits, Western Australia, Aust. J. Earth Sci., 56(8), 1111–1142, doi:10.1080/08120090903246220, 2009.

Yamamoto, J. K., Koike, K., Kikuda, A. T., Campanha, G. A. da C. and Endlen, A.: Post-processing for uncertainty reduction in computed 3D geological models, Tectonophysics, 633(1), 232–245, doi:10.1016/j.tecto.2014.07.013, 2014.

Yan, P., Kalscheuer, T., Hedin, P. and Garcia Juanatey, M. A.: Two-dimensional magnetotelluric inversion using reflection seismic data as constraints and application in the COSC project, Geophys. Res. Lett., 44(8), 3554–3563, doi:10.1002/2017GL072953, 2017.

Zhdanov, M. S., Gribenko, A. and Wilson, G.: Generalized joint inversion of multimodal geophysical data using Gramian constraints, Geophys. Res. Lett., 39(9), L09301, doi:10.1029/2012GL051233, 2012.

Summary
5/11/2018 8:58:59 PM

Differences exist between documents.

**New Document:**
20181105_ManuscriptRev
27 pages (1.71 MB)
5/11/2018 8:58:28 PM
Used to display results.

**Old Document:**
2018824_ManuscriptUpload
22 pages (1.39 MB)
5/11/2018 8:58:27 PM

Get started: first change is on page 1.

No pages were deleted

**How to read this report**

Highlight indicates a change.
Deleted indicates deleted content.
▲ indicates pages were changed.
⬌ indicates pages were moved.

[revised manuscript text omitted]
}(\boldsymbol{d}-\boldsymbol{g}(\boldsymbol{m}))\right\|_2^2}_{\text{Data term}} + \underbrace{\alpha_m\left\|\boldsymbol{W_m}(\boldsymbol{m}-\boldsymbol{m_p})\right\|_2^2}_{\text{Model term}} + \underbrace{\alpha_H\left\|\boldsymbol{W_H}\nabla\boldsymbol{m}\right\|_2^2}_{\text{Structural regularization term}}, \quad (1)$$

where $\boldsymbol{d}$ relates to the geophysical measurements to be inverted, $\boldsymbol{g}$ is the forward modelling operator; $\boldsymbol{m}$ relates to the model being searched for, and $\boldsymbol{m_p}$ is the prior model; $\boldsymbol{W_d}$, $\boldsymbol{W_m}$ and $\boldsymbol{W_H}$ are diagonal weighting matrices corresponding to data noise, model weighting and gradient regularization, respectively. The model term is a ridge regression constraint term (Hoerl and Kennard, 1970).

The structural regularization term in Eq. (1) enforces structural constraints during inversion. It is weighted locally by matrix $\boldsymbol{W_H}$, which can be derived from prior information (see Subsect. 2.3 for details). The positive free parameters $\alpha_m$ and $\alpha_H$ control the overall weight of model and structural regularization terms; $\nabla$ is the spatial gradient operator. Note that $\|\nabla\boldsymbol{m}\|_2$, estimates the amount of structure in inverted physical property model $\boldsymbol{m}$. Also note that parts of the model where $\boldsymbol{W_H} = 0$ are excluded from the calculation of the structural regularization and can change freely to accommodate geophysical data.

We utilize the integrated sensitivities technique of (Portniaguine and Zhdanov, 2002) to balance the decreasing sensitivity of gravity data with depth. We chose this technique because it offers the advantage of providing "equal sensitivity of the observed data to the cells located at different depths and at different horizontal positions" (Vatankhah and Renaut, 2017).

**2.2 Probabilistic geological modelling**

Probabilistic geological modelling is performed using the Monte-Carlo Uncertainty Estimator (MCUE) method of (Pakyuz-Charrier et al., 2018b, 2018a), which extends previous works from (Jessell et al., 2010; Lindsay et al., 2012; Wellmann et al., 2010). It is a 3D uncertainty propagation method for implicit geological modelling that uses geological rules and geological orientation measurements (foliation and interface of the geological units sampled at surface level or in borehole) as inputs. The sampling algorithm perturbs orientation data used to derive a reference model by sampling probability distributions describing the uncertainty of orientation data to produce a series of unique altered geological models. Realizations that do not respect a series of geological rules are considered implausible and are rejected. Coupled to the 3D geological modelling engine of Geomodeller© (Calcagno et al., 2008), it produces a set of plausible geological models honouring the geological input measurements that represent the geological model space (Lindsay et al., 2013b). The observation probabilities constituting the probabilistic geological model (PGM) are obtained, in each model cell, by calculating the relative observation frequencies of the different lithologies from the set of geological models. For the $i^{th}$ model cell of a PGM containing $L$ lithologies, vector $\boldsymbol{\psi^i} = \left[\psi^i_{k=1}, \dots, \psi^i_{k=L}\right]$ contains the observation probabilities of each lithology. As we show in the next subsection, the resulting PGM can be used to estimate uncertainty levels and as a source of prior information.

**2.3 Utilisation of information entropy for local conditioning**

Information entropy was introduced for geological modelling by Wellmann and Regenauer-Lieb (2012) and is gaining popularity as a measure of uncertainty in probabilistic geological modelling (de la Varga et al., 2018; de la Varga and Wellmann, 2016; Lindsay et al., 2014; Lindsay et al., 2013; Pakyuz-Charrier et al., 2018; Schweizer et al., 2017; Thiele et al., 2016; Wellmann et al., 2017; Yamamoto et al., 2014). Quoting (Schweizer et al., 2017), information entropy "quantifies the amount of missing information and hence, the uncertainty at a discrete location". For the $i^{th}$ model-cell, it is given as (Shannon, 1948):

$$H^i = H\left(\boldsymbol{\psi^i}\right) = -\sum_{k=1}^{L} \psi^i_k \log\left(\psi^i_k\right). \tag{2}$$

Fundamentally, geological uncertainty contained in $\boldsymbol{H}$ encapsulates information about possible locations of interfaces between units and areas where the geological data is insufficiently informative. Instead of using $\boldsymbol{H}$ directly, we calculate $\boldsymbol{W_H}$ utilising

its normalized complementary, which reflects the degree of certainty across the model. Let us express $W_H$ as follows, for the $i^{th}$ model cell:

$$W_H^i = \frac{\max \boldsymbol{H} - H^i}{\max \boldsymbol{H} - \min \boldsymbol{H}} \tag{3}$$

The consequence of Eq. (2) and 3 is that $W_H$ is minimum at interfaces and in areas poorly constrained by geological information, and equal to unity in areas where the geology is well resolved. Consequently, the conditioning process attaches small weights to the structural term of Eq. (1) in uncertain cells, while consistently high values will be applied to the most geologically certain cells. As a result, it enables the inversion algorithm to favour nearly constant changes in the inverted model in contiguous certain groups of cells (e.g., where $W_H \rightarrow 1$) while relaxing the regularisation constraints in the most uncertain parts (e.g., where $W_H \rightarrow 0$).

**3     Proof of concept: synthetic case study**

This section introduces the proof of concept of the proposed method through an idealized case study illustrating the potential of the proposed inverse modelling scheme to ameliorate inversion results and to reduce interpretation uncertainty. We use the same 3D density contrast model as (Giraud et al., 2017), which is obtained by populating the structural framework of (Pakyuz-Charrier et al., 2018b). We simulate a series of PGM sought to represent expected values as well as possible extreme scenarii. The short presentation of the model below and the analysis of results provides essential information about the synthetic survey and shows the proof of concept of the methodology used in the paper.

**3.1     Survey setup**

The 3D unperturbed reference geological model was generated from contact (interface points) and surface orientation (foliations) field measurements collected in the Mansfield area (Victoria, Australia). It presents a Carboniferous mudstone-sandstone basin oriented N170, abutting a faulted contact with a folded ultramafic basement to the South-West. Model complexity was artificially increased through the addition of a North-South fault and of a mafic intrusion.

The true density contrast model (Figure 1a) was obtained assigning density contrasts consistently with the structural setting of the reference geological model, assuming a flat topography. Density contrasts of 0 and 100 kg.m$^{-3}$ were assigned to the upper and lower basin lithotypes, respectively. Mafic rocks were assigned a density contrast of 200 kg.m$^{-3}$ while the density contrast of the ultramafic basement was set to 300 kg.m$^{-3}$.

MCUE perturbations of the reference geological model were first performed using standard measurement uncertainty values recommended by metrological studies as reported by (Allmendinger et al., 2017; Novakova and Pavlis, 2017). We generated a series of 300 models that were subsequently combined into a PGM. The resulting volume representing the $W_H$ values calculated from this PGM in each cell of the model as per Eq. (3) is shown in Figure 1b.

[Figure]

**Figure 1.** True density contrast model and $W_H$ values used for local regularization conditioning. (a) Unperturbed reference model populated with density contrast values, (b) uncertainty values used for local regularization conditioning.

**3.2 Locally constrained inversion: validation**

5 To assess the impact of local conditioning of the regularization function onto inversion, we compare inversions using non-conditioned (Figure 2a) and locally conditioned (Figure 2b) regularization function, respectively. Please note that, simulating the absence of prior petrophysical information, a homogenous prior model set to 0 kg.m⁻³ was used in both cases.

[Figure]

**Figure 2**. Comparison of inversion results. (a) Inverted models with non-conditioned regularization weights, and (b) using local conditioning.

Besides qualitative visual comparison of the models, we interpret inversion results (Figure 2) through the commonly used model and data root-mean-square error (RMSE), which correspond to the model and data terms calculated with weights and
5  covariances set to unity. We evaluate the geometrical similarity between inverted and true model through the Bravais-Pearson correlation (also often called 'linear correlation coefficient') between their gradients (Table 1).

Comparison of the true model (Figure 1a) with inversion results in Figure 2a and Figure 2b shows that while the structures in shallowest part of the model are well retrieved in both cases, it appears that they are considerably better recovered through usage of conditioned regularization overall (Figure 2b). The guiding effect of $W_H$ is visible in Figure 2b where the main
10  structures at depth follow the general features of the conditioning volume (Figure 1b). Moreover, in order to minimize the conditioned regularization constraint simultaneously to data misfit, inversion was driven to accommodate inverted model values (Figure 2b) such that they are closer to the causative model (Figure 1a) than without conditioning (Figure 2a). This leads to reduced model RMSE on the one hand, and data RMSE on the other (Table 1). This reduction in data RMSE can also be explained by the relaxation of the constraints in several portions of the model, thus increasing the degree of liberty to
15  accommodate the model towards lower data misfit. More importantly, the Bravais-Pearson correlation between the inverted and true model gradients is much higher when information from information entropy is used. This indicates that local

conditioning of the regularization function also allows for significantly better retrieval of the causative bodies' (e.g., the true model) structural features.

Please also note that we do not show the recovered geophysical measurements because visual differences between recovered and inverted measurements are minimal.

**Table 1.** Indicators for comparison of inversion results in terms of model, data, and structure.

| | Model RMSE (kg.m⁻³) | Data RMSE (m.s⁻²) | Correlation between gradients |
|---|---|---|---|
| *Non-conditioned regularization* | 74.66 | $2.38 \times 10^{-9}$ | 0.18 |
| *Locally conditioned regularization* | 53.05 | $7.44 \times 10^{-10}$ | 0.53 |

From these observations, we conclude that the application of the local conditioning scheme can fulfil the objectives of data integration in inversion as it allows inversion to recover models that are closer to the causative bodies and easier to interpret, while honouring geophysical data. Nevertheless, it remains important to test the methodology in cases where the uncertainty indicator $W_H$ is biased and/or shows high geological uncertainty levels away from faults and contacts. A thorough analysis lying beyond the scope of this paper, the remainder of this section presents an elementary sensitivity analysis using a series of two $W_H$ volumes representing distinct extreme scenarios.

**3.3 Inversion constrained by biased geological uncertainty model**

In this subsection, we investigate the effect of inaccurate geological models and the propagation of the related uncertainty in inversion. For this purpose, we calculate a second PGM from MCUE perturbations in which we split the ultramafic basement into two independent units, without changing the density contrast values (Figure 3a). This results in the existence of a fictitious geological unit that is invisible to gravity data and presents no density contrast but which increases geological complexity and uncertainty (Figure 3b) (we further refer to it as 'ghost' geological unit). Notably, it increases geological uncertainty and smears interfaces that are well-constrained as per Figure 1. It also decreases $W_H$ in large parts of the model where $W_H \rightarrow 1$ previously, thereby favouring model changes in these areas during inversion and encouraging it to place larger density contrast or interfaces in these areas.

[Figure]

**Figure 3.** (a) true density contrast model with outline of the 'ghost' unit B (black dashed line), embedded in ultramafic unit A, and (b) local weights calculated from the PGM calculated using MCUE for model (a).

5   Comparison of the inverted models obtained without (Figure 4a) and with the ghost unit (Figure 4b) reveals that they exhibit broadly similar features except in the most geologically complex parts of the model as per Figure 1b, where differences are minor. This indicates that while geophysical inversion updates the prior density contrast model preferably in geologically uncertain regions, low values of $W_H$ do not necessarily lead to the modelling of an interpretable interface by inversion. From the comparison of Figure 3b and Figure 4b, we can deduce that local conditioning as applied in this work does not necessarily
10   enforce strict structural similarity between the inverted model and the conditioning geological uncertainty volume.

[Figure]

**Figure 4.** Comparison of inverted model using $W_H$ derived from a PGM considering the ghost unit (b) and without it (a); (a) is the inverse model obtained without bias in the PGM as per Figure 2 and is shown here for comparison with (b).

**3.4    Inversion constrained using exaggerated geological uncertainty**

5    To complete this series of tests, we generated a third PGM showing exaggerated geological uncertainty. To this end, we used a half aperture 95% confidence interval of ~50 degrees for orientation data measurements in our MCUE simulations. This is far higher than for the rest of the MCUE simulations used in this paper. All other simulations (Subsect. 3.2 and 3.3) use a value of ~11 degrees, which is representative of realistic measurement uncertainty as proposed by recent metrological studies (Allmendinger et al., 2017; Cawood et al., 2017; Novakova and Pavlis, 2017). Figure 5 below shows the resulting $W_H$ volume

10    (Figure 5a) and the inverted model obtained using it for local conditioning of the regularization constraint (Figure 5b).

[Figure]

**Figure 5.** (a) Local weights $W_H$ calculated from a PGM corresponding to exaggerated uncertainty in geological input measurements and (b) corresponding inverted density contrast model.

The features visible in Figure 5a reflect the high geological input measurement uncertainty. Geologically uncertain areas cover large portions of the volume and only the simplest geological structures (e.g., the basin) seem to be well constrained by geology. Areas of the model previously well constrained (Figure 1b) present varying degrees of uncertainty. This illustrates that, as can be expected, increasing geological input uncertainty translates in relaxing the guiding effect of local conditioning using $W_H$, which results in geophysical inversion being less strongly guided by geological information. As can be seen in Fig 4b, the inverted model obtained in this case shows structures that present weaker contrast around interfaces than when geological uncertainty is lower (Figure 2b). Importantly, however, most structures are well preserved and the overall model values for the different lithologies remain closer to the true model than for the non-conditioned case (Figure 2a). This indicates that even in high geological uncertainty scenarios, interpretation outcome may be largely more reliable when local regularisation is used.

The analysis and comparison of the results shown in this section demonstrates the potential of the proposed inverse modelling scheme to ameliorate inversion results and to reduce interpretation uncertainty. It also illustrates the capability of our methodology to deal with high or biased conditioning uncertainty estimates. In this synthetic case, local conditioning allows

geophysical inversion to significantly improve the imaging of geologically uncertain areas and to exploit complementarities between geological modelling and geophysical inversion. From the success of this proof of concept study, we are confident that our integration method can be tested here using real world data for field validation.

**4    Field validation: Yerrida Basin case study**

**4.1    Geological context and geophysical survey setup**

The Yerrida Basin is located in the southern part of the Capricorn Orogen, at the northern margin of the Yilgarn Craton in Western Australia (Figure 6a), and extends approximately 150km N-S and 180 km E-W (Figure 6b). The studied area is bounded in the northwest by the Goodin Fault, which represents a faulted contact between the Yerrida Basin and the Bryah-Padbury Basin. The structures of interest in this work are the Archean greenstone belts extending north-northwest that are unconformably overlain by Paleoproterozoic sedimentary rocks the form the Yerrida Basin. Features A and B (Figure 6a and Figure 6b) indicate the interpreted position of buried Wiluna Greenstone Belt. Where the Wiluna Greenstone Belt is exposed, it is host to base and precious metal mineralisation (Williams, 2009). With a relatively high positive density contrast (expected to be between 190 and 270 kg.m$^{-3}$) to the Yilgarn Craton granite-gneiss host, mafic greenstone belts A, B, and C are suitable targets for gravity inversion. Interpretations from multiple studies in the region, e.g, (Johnson et al., 2013; Pirajno et al., 1998; Pirajno and Adamides, 2000; Pirajno and Occhipinti, 2000) were compiled while additional geological measurements acquired in 2015, 2016 and 2017 complemented legacy data (Occhipinti et al., 2017; Olierook et al., 2018). This allowed the revision of existing models and improved our understanding of the area. This, in turn, also highlights the challenges presented by the characterization of greenstone belts A, B and C, and that further geophysical analysis through constrained inversion is a useful pathway for reducing exploration risk.

Geophysical data consists of ground surveys obtained from Geoscience Australia (http://www.ga.gov.au/data-pubs). Post-processing includes spherical-cap and terrain corrections and the removal of the regional trend to obtain the complete Bouguer anomaly. As most data points were acquired 1 to 4 km apart, the dataset was resampled to match the geological model discretization, making up a total of 4882 measurement points. The model is discretized into $100 \times 100 \times 42$ cells of dimensions 2.335 km $\times$ 1.875 km $\times$ 1.0475 km, down to a depth of 44 km, making up a total of 420000 cells.

[Figure]

**Figure 6.** Geological context and geophysical data. (a) Geological map of the area and (b) complete Bouguer anomaly. The dashed lines delineate the possible sub-basin extent of the mafic greenstone belts A, B and C.

**4.2 Geological modelling**

Geological information consists of in-situ structural measurements (interfaces and foliations) and interpretation of aeromagnetic, airborne electromagnetic, Landsat 8 and ASTER hyperspectral data. Legacy data from the Geological Survey of Western Australia (Pirajno and Adamides, 2000) and CSIRO (Ley-Cooper et al., 2017) were used, to which about 600 surface geological and petrophysical measurements from recent geological field campaigns were added. Although the available petrophysical measurements were not used to derive petrophysical constraints because of the insufficient sampling of several of the modelled lithologies, they were a useful source of information to populate geological models and for interpretation. Remote-sensing data were used to test interpretations.

These datasets were used jointly to build a reference geological model reconciling the available geological information in Geomodeller.

Lithologies with similar density contrasts were merged and subsequently treated as a single rock type in MCUE simulations. Uncertainty related to structural measurements was subsequently used as inputs to the MCUE perturbations (Pakyuz-Charrier et al., 2018b) of the reference model to generate a collection of 500 accepted models. Information extracted from the PGM is displayed in Figure 7. Figure 7a shows the lithologies with the highest probability for each cell of the PGM. The associated $W_H$ values used in inversion are shown in Figure 7b. The prior model for inversion $m_p$ is equal to the mean model of the 500 plausible models generated by MCUE. It is shown in Figure 7c.

[Figure]

**Figure 7.** Geological modelling results. (a) Most probable lithology in each model cell (same colour code as in Figure 6) (b) values used for local regularization conditioning, (c) and prior model derived from PGM and prior petrophysical information). In (a), "background" refers to all the lithologies that have a density contrast equal to 0 kg.m⁻³.

**4.3 Inversion results and analysis**

The aim of our analysis is to assess the influence of the local conditioning of structural constraints on inversion through comparison with the non-conditioned case, all other things remaining constant.

**4.3.1 Comparative analysis strategy**

5 Prior to examination of the inverted models, we analyse geophysical data misfit after inversion. This enables us to ensure that the inversion results we compare produce, in our case, similar gravity anomalies. Our study of inverted models focuses on results obtained through usage of non-conditioned (Figure 8a) and conditioned regularization function (Figure 8b) using $W_H$ (Figure 7b). In addition to departures from the prior 
[revised manuscript text omitted]

The quantitative integration technique we presented reduces uncertainty and ambiguity compared to qualitative interpretation

15  technique or single-discipline workflows. However, despite its robustness to misplaced interface (e.g., bias) or to high geological uncertainty (e.g., sparse or very uncertain geological input measurements) as shown in the synthetic case, interpreters need to bear in mind the specificities of the geophysical data inverted for (resolution of specific geometries, depth of investigation) and the shortcomings of geological modelling workflows. As for all geological modelling, MCUE is oblivious to geological units or faults that are not sampled by field geological measurements, which can lead to biases in final models

20  due to, for instance, inclusions not be accounted for.

Current research comprises the development of sensitivity and resolution analyses in an effort to mitigate the risk of the model being affected by uncertainty sources not accounted for. Future research will include the utilization of local petrophysical constraints of (Giraud et al., 2017) in the uncertainty-guided inversion scheme we presented, as well as the utilisation of geological uncertainty to weight the cross-gradient term of (Gallardo and Meju, 2003) locally. With this last respect,

25  uncertainty-guided inversion can be assisted in the most uncertain parts of the model by guided inversion (in the sense of Brown et al., 2012) or through cross-gradient joint inversion.

*Code and data availability.* True property models, inversion results and recovered models relating to the Yerrida Basin shown in this article are made available online: Jeremie Giraud, Mark Lindsay, and Vitaliy Ogarko, 2018, Yerrida Basin Geophysical

30  Modeling - Input data and inverted models. (Version version 1.0) [Data set]. Zenodo. http://doi.org/10.5281/zenodo.1238216. True property models, inversion results and recovered models relating to the synthetic case from the Mansfield area shown in this article are made available online: Jeremie Giraud, Vitaliy Ogarko, and Evren Pakyuz-Charrier, 2018, Synthetic dataset for

the testing of local conditioning of regularization function using geological uncertainty. (Version version 1.0) [Data set]. Zenodo. http://doi.org/10.5281/zenodo.1238529

**Appendix A: Data misfit maps from inversion in the Yerrida Basin**

Figure A below relates to the analysis of data misfit in Sect. 4 of the article through the plot of the data misfit maps for the

5 non-conditioned and conditioned cases (Fig. Ad and Fig. Ah, respectively). It is complemented by the corresponding plots of starting (Fig. Aa and Fig. Ae), observed (Fig. Ab and Fig. Af), and calculated data (Fig. Ac and Fig. Ah). Note that Fig. Ac and Fig. Ag show little visual differences, and that Fig. Ad and Fig. Ah exhibit similar features while showing limited coherent signal.

[Figure]

**Figure A.** Comparison of input and output geophysical data. (a) and (e) show data calculated from the prior model, (b) and (f) input

measurements, (c) and (g) data calculated from the inverted model, and (d) and (f) the absolute value of the difference of the misfit between the observed and calculated data. (a)-(d) (i.e., first line) and (e)-(h) (i.e., second line) correspond to the non-conditioned and conditioned cases, respectively.

15 *Authors contribution.* Jeremie Giraud performed the integrated inverse modelling of geophysical data for both the Mansfield synthetic study and the Yerrida Basin. He performed posterior analysis and interpretation of results and he is the main contributor to the writing of this article. Mark Lindsay acquired part of the geological field measurements from the Yerrida Basin and performed the geological modelling of the area. He participated in the writing of the geological setting subsection and he produced the geological map shown in Figure 6a. Vitaliy Ogarko and Jeremie Giraud worked together on the

20 implementation and testing of the proposed methodology in Tomofast-x, of which Vitaliy Ogarko, Roland Martin and Jeremie Giraud are the main developers. Mark Jessell has been involved in the validation of the methodology at the initial development

stage and supervised the progress of the presented work. Roland Martin provided support at the initial stage of the inversion of gravity data from the Yerrida Basin. Evren Pakyuz-Charrier assisted Mark Lindsay with the utilisation of MCUE. All co-authors contributed to the final version of this article. Mark Lindsay and Vitaliy Ogarko were the most actively involved in the revision process of the drafts leading to this paper.

5 *Competing interests*. The authors declare that they have no conflict of interest.

*Acknowledgements*. Appreciation is expressed to the CALMIP supercomputing centre (Toulouse, France), for their support through Roland Martin's computing projects #P1138_2017 and #P1138_2018 and for the computing time provided on the EOS machine. Jeremie Giraud is a recipient of the International Postgraduate Research Scholarship from the Australian Federal Government and he received a grant from the Australian Society of Exploration Geophysics Research Foundation. The authors
10 also thank Peter Lelievre for interesting discussions and constructive feedback relating to the utilisation of gradient-related constraints. Mark D. Lindsay and Mark W. Jessell thank the Geological Survey of Western Australia (Royalties for Regions Exploration Incentive Scheme) and the Minerals Research Institute of Western Australia for their support. Mark W. Jessell is supported by a Western Australian Fellowship. The authors acknowledge use the Zenodo research date repository to share the data necessary to reproduce the presented work. They finally thank Colin Farquharson and an anonymous reviewer for their
15 review of the manuscript.

---

## Referee Report (RR1)

[Figure]

*Figure 1. (a) true density contrast model with outline of the 'ghost' unit B (black dashed line), embedded in ultramafic unit A, and (b) local weights calculated from the PGM calculated using MCUE for model (a).*

[Figure]

5